# Fairer AI in ophthalmology via implicit fairness learning for mitigating sexism and ageism

Weimin Tan [1,3], Qiaoling Wei [2,3], Zhen Xing [1], Hao Fu[1], Hongyu Kong[2], Yi Lu[2] ✉, Bo Yan [1] ✉ & Chen Zhao [2] ✉

The transformative role of artificial intelligence (AI) in various fields highlights the need for it to be both accurate and fair. Biased medical AI systems pose significant potential risks to achieving fair and equitable healthcare. Here, we show an implicit fairness learning approach to build a fairer ophthalmology AI (called FairerOPTH) that mitigates sex (biological attribute) and age biases in AI diagnosis of eye diseases. Specifically, FairerOPTH incorporates the causal relationship between fundus features and eye diseases, which is relatively independent of sensitive attributes such as race, sex, and age. We demonstrate on a large and diverse collected dataset that FairerOPTH significantly outperforms several state-of-the-art approaches in terms of diagnostic accuracy and fairness for 38 eye diseases in ultra-widefield imaging and 16 eye diseases in narrow-angle imaging. This work demonstrates the significant potential of implicit fairness learning in promoting equitable treatment for patients regardless of their sex or age.

Fairness in artificial intelligence (AI) has attracted great attention recently because AI has penetrated into every aspect of our lives. AI models have the risk of over-associating sensitive attributes such as age, race, and sex with the decision-making process[1,2]. These models may exhibit discriminatory behavior against specific subgroups. In the context of fundus disease diagnosis, unfairness refers to bias or favoritism against individuals or subgroups of a specific age or sex, i.e., sexism or ageism in AI diagnostic models. Fairness issues may have serious adverse effects on individuals and society and further exacerbate social inequality[3].

Sexism and ageism are pervasive biases in various healthcare disciplines, including the field of ophthalmology. These biases can lead to disparities in diagnosis, treatment, and care, compromising patient outcomes and equity. By developing a fairer AI for applications in ophthalmology, we can mitigate these biases and provide fair and unbiased eye care[4,5]. Despite the remarkable diagnostic accuracy of data-driven deep neural networks (DNNs), DNNs are susceptible to bias due to the prioritization of minimizing overall prediction errors,

ultimately leading to unfair decisions[6]. For instance, studies have highlighted the biases of automated diagnostic systems, wherein individuals from certain racial, sex, or age groups may receive more accurate diagnostic results. These biases have caused great concern to society and government[7,8]. Alarmingly, unfair AI diagnostic systems may further amplify social inequity and unfairness, because the unfair AI diagnoses may affect subsequent diagnosis and treatment decisions, causing certain groups to face a higher rate of misdiagnosis and receive, delayed or unnecessary treatment. This vicious cycle further leads to greater data bias, perpetuating and exacerbating inequality and injustice[9–11].

The field of ophthalmology faces significant challenges in achieving AI fairness[6]. These challenges arise from several key factors. First, sample imbalance presents an obstacle, as different ophthalmic diseases have different prevalence, resulting in uneven distributions of the samples used during algorithm training and evaluation. This imbalance can lead to biased performance, with common diseases receiving more attention than rare ones. Second, the inherent diversity

[1]School of Computer Science, Shanghai Key Laboratory of Intelligent Information Processing, Fudan University, Shanghai, China. [2]Eye Institute and Department of Ophthalmology, Eye & ENT Hospital, Fudan University, Shanghai, China. [3]These authors contributed equally: Weimin Tan, Qiaoling Wei. ✉e-mail: luyieent@126.com; byan@fudan.edu.cn; dr_zhaochen@fudan.edu.cn

within retinal fundus images adds complexity to the issue of achieving fairness. Fundus images can show a wide range of ophthalmic diseases, each with distinct characteristics and manifestations. Developing algorithms that can effectively address the diversity across and within disease groups are crucial for fair and accurate diagnosis.

Furthermore, disparities associated with individual attributes, such as age, sex, region, and race, pose additional challenges[12,13]. Variations in the incidence and prevalence of ophthalmic diseases among different demographic groups introduce the potential for biased predictions and unequal outcomes. Ensuring fair performance across diverse populations requires careful consideration and appropriate mitigation strategies. Finally, data biases can undermine fairness in ophthalmology. Biases may arise from the data collection methods and data sources, leading to under-representation or over-representation of certain populations or geographic regions. Such biases can limit the generalizability and fairness of algorithms in real-world applications. Addressing these challenges requires the development of fairness-aware algorithms in ophthalmology that account for sample imbalances, accommodate disease diversity, mitigate attribute biases, and use data obtained with fair data collection practices. By promoting fairness, the field of ophthalmology can advance towards more equitable and unbiased healthcare delivery.

In this study, we propose an implicit fairness learning approach to build a fairer ophthalmology AI (called FairerOPTH) that mitigates sex (biological attribute) and age biases in AI diagnosis of eye diseases. Specifically, FairerOPTH incorporates the causal relationship between fundus features and eye diseases, which is relatively independent of sensitive attributes such as race, sex, and age (Supplementary Fig. 1). Our results show that the proposed FairerOPTH can more fairly and precisely screen disease presence from fundus images. The casual relationship between fundus features and disease diagnosis approximates the clinical diagnosis process and helps AI models to deal with the complex diseases with the changing of fundus features. Therefore, using this causality as prior knowledge for AI models can encourage models to learn representations that are relatively independent to sensitive attributes, thereby enabling unbiased classification.

To evaluate the FairerOPTH, we collected a large and diverse fundus dataset from over 8,405 patients across a wide age range (0–90 years). The fundus dataset contains advanced ultra-widefield (UWF) and regular narrow-angle (NAF) fundus imaging datasets (Fig. 1 and Supplementary Table 1). These two new datasets have unique advantages with respect to the labeling of fundus features compared with public fundus datasets. The collected UWF dataset called OculoScope, a comprehensive fundus imaging dataset, contains 16,530 UWF images, 38 ophthalmic diseases, and 67 fundus features. The NAF dataset called MixNAF, which is compiled from four public NAF datasets and our newly collected NAF images, contains 4540 NAF images, representing 20 ophthalmic diseases and 16 fundus features. Both the OculoScope and MixNAF datasets are used to evaluate the fairness and accuracy of FairerOPTH. Our results show that the implicit fairness learning method can significantly mitigate unfairness in terms of sexism and ageism while improving the accuracy of the multi-disease classification and this method merits further study.

## Results

We develop and validate a fairer AI approach for ophthalmology (FairerOPTH) using OculoScope and MixNAF datasets containing data from over 8,405 patients. We adopt six fairness-based metrics including $\Delta D$ (screening quality disparity), $\Delta M$ (max screening disparity), $\Delta A$ (average screening disparity), PQD (predictive quality disparity), DPM (demographic disparity metric), and EOM (equality of opportunity metric)[2] (see 'Fairness evaluation metrics' Section in Methods) and four commonly used screening accuracy metrics including mAP (mean average precision), specificity, sensitivity, and AUC (area under the curve) to evaluate FairerOPTH. Specifically, we first conduct

extensive experiments with the OculoScope dataset to evaluate the fairness of FairerOPTH in terms of two sensitive attributes, age and sex. Then, we evaluate the diagnostic accuracy of FairerOPTH with the OculoScope and MixNAF datasets and show the advanced performance of FairerOPTH by comparing it with state-of-the-art methods. Finally, we conduct ablation studies to verify the effectiveness of the proposed pathology-aware attention module and different loss terms.

### Mitigating ageism

We first evaluate FairerOPTH for its capability to fairly screen different age groups of patients (Fig. 2). Across the four age divisions, comprising patients groups up to 40 years, FairerOPTH achieves an average screening disparity, $\Delta A$ of 31.9%, 29.3%, 15.4%, and 12.4%, respectively (Fig. 2b) and a max screening disparity $\Delta M$ of 122.7%, 122.7%, 62.7%, and 154.6%, respectively. The multi-label classification model using ResNet-101 as the backbone (baseline model) achieves a $\Delta A$ of 44.3%, 38.4%, 24.4%, and 14.8% and a $\Delta M$ of 131.8%, 134.4%, 105.3%, and 185.0%, respectively. Compared with the baseline model, FairerOPTH has significantly mitigated the age bias when screening 38 diseases from UWF images. The DPM and EOM also demonstrate the superior performance of FairerOPTH. The $\Delta D$ and PQD of different age groups of each disease are shown in Fig. 2c, d in detail. In most cases, $\Delta D$ and PQD of FairerOPTH is better than that of the baseline model for the 38 diseases. We also show the comparison of the screening accuracy of each disease corresponding to the patient 10-year age groups (Table 1).

### Mitigating sexism

Next, we evaluated the efficacy of FairerOPTH for the equitable screening of ocular diseases in patients of different sexes. On the OculoScope test set, the FairerOPTH achieves a mAP of 91.8% and 90.8% for females and males, respectively, while the baseline model achieves values of 88.1% and 86.8% for females and males, respectively (Fig. 3). For females and males, the average screening accuracy of FairerOPTH is increased by 3.7% and 4.0% mAP over the baseline performance, respectively (Fig. 3b). In addition, FairerOPTH achieves a $\Delta A$ of 7.6%, a $\Delta M$ of 66.5%, a DPM of 82.4, and an EOM of 97.6, while the baseline model achieves a $\Delta A$ of 10.3%, a $\Delta M$ of 60.5%, a DPM of 71.2, and an EOM of 96.1. FairerOPTH has significant advantages in terms of lower average screening disparity, DPM, and EOM. The $\Delta D$ and PQD of FairerOPTH for each disease are shown in Fig. 3c, d. Compared with the baseline model, FairerOPTH has smaller $\Delta D$ values for 38 diseases, which means that FairerOPTH is fairer when considering the sex of patients, showing less sex bias, especially for diagnosing lens dislocation. For myelinated nerve fiber, optic abnormalities, and other diseases, FairerOPTH also shows large fairness advantages. Finally, we report the screening accuracy (mAP) for each disease in patients of different sexes (Table 2). FairerOPTH can make diagnoses that are not only less biased but also more accurate. Even when screening 38 disease types with a significantly imbalanced distribution (Fig. 1), we show that FairerOPTH still demonstrates great improvements in accuracy and fairness over the baseline model.

### Screening accuracy on the OculoScope dataset

To verify that FairerOPTH is not solely able to mitigate unfairness, we examine its multitask multilabel classification accuracy on UWF and NAF fundus images to understand the effectiveness of incorporating fundus features into the disease diagnosis process (Fig. 4a). We compare FairerOPTH with several state-of-the-art multi-label classification approaches, including the MCAR[14], CSRA[15], C-Tran[16], ML-Decoder[17], and ASL[18] approaches.

The approach for each comparison is slightly modified to identify the 38 diseases included in the OculoScope dataset. The comparison results evaluated on the OculoScope dataset are shown in Fig. 4b. The performance of the MCAR[14] method is relatively poor because it uses a

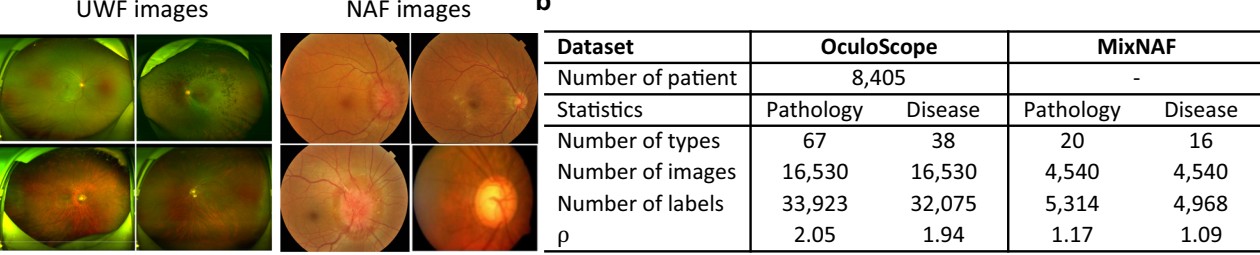

**Fig. 1 | Characteristics of the datasets. a** Fundus images including both ultra-widefield (UWF) and narrow-angle (NAF) images, play a foundation role in the diagnosis of various ocular diseases by modern clinical systems. UWF imaging, an advanced method, covers up to 200° eccentricities in a single capture, while NAF imaging, which is more common, has an angle of view of 30–60°. **b** Statistics of the OculoScope and MixNAF datasets. The label density $\rho = \frac{1}{N}\sum_{i=1}^{K}|y_i|$ quantitatively shows that a single fundus map contains multiple diseases and pathological signs on average, where $y_i$ is the number of the $i$-th disease, $N$ represents the total number of images, and $K$ represents the number of disease types. **c** The sample distribution of 38 diseases in OculoScope is extremely unbalanced. **d** The sample distribution of 16 diseases on MixNAF is also extremely unbalanced. **e** Visual representation of fundus features in fundus images. Source data are provided as a Source Data file.

CNN framework and does not consider the problem of class imbalance. Therefore, this method is not suitable for images with scattered pathological features such as fundus images. Likewise, although the CSRA[15] method shows a slight improvement, this improvement may be more obvious on images with more concentrated features such as natural images. For fundus images with widely distributed and scattered fundus features of the same category, C-Tran[16] and ML-Decoder[17] use a transformer-based classification head, and their performance is improved to a certain extent. ASL[18] somewhat addressed the class imbalance problem, so its performance is improved significantly. Our FairerOPTH method achieves the best results compared to the above methods. The mAP of FairerOPTH is 92.0%, which is 2.2% larger than that of ASL[18]. FairerOPTH achieves the sensitivity and specificity of 0.956 and 0.970, respectively. In summary, the experimental results

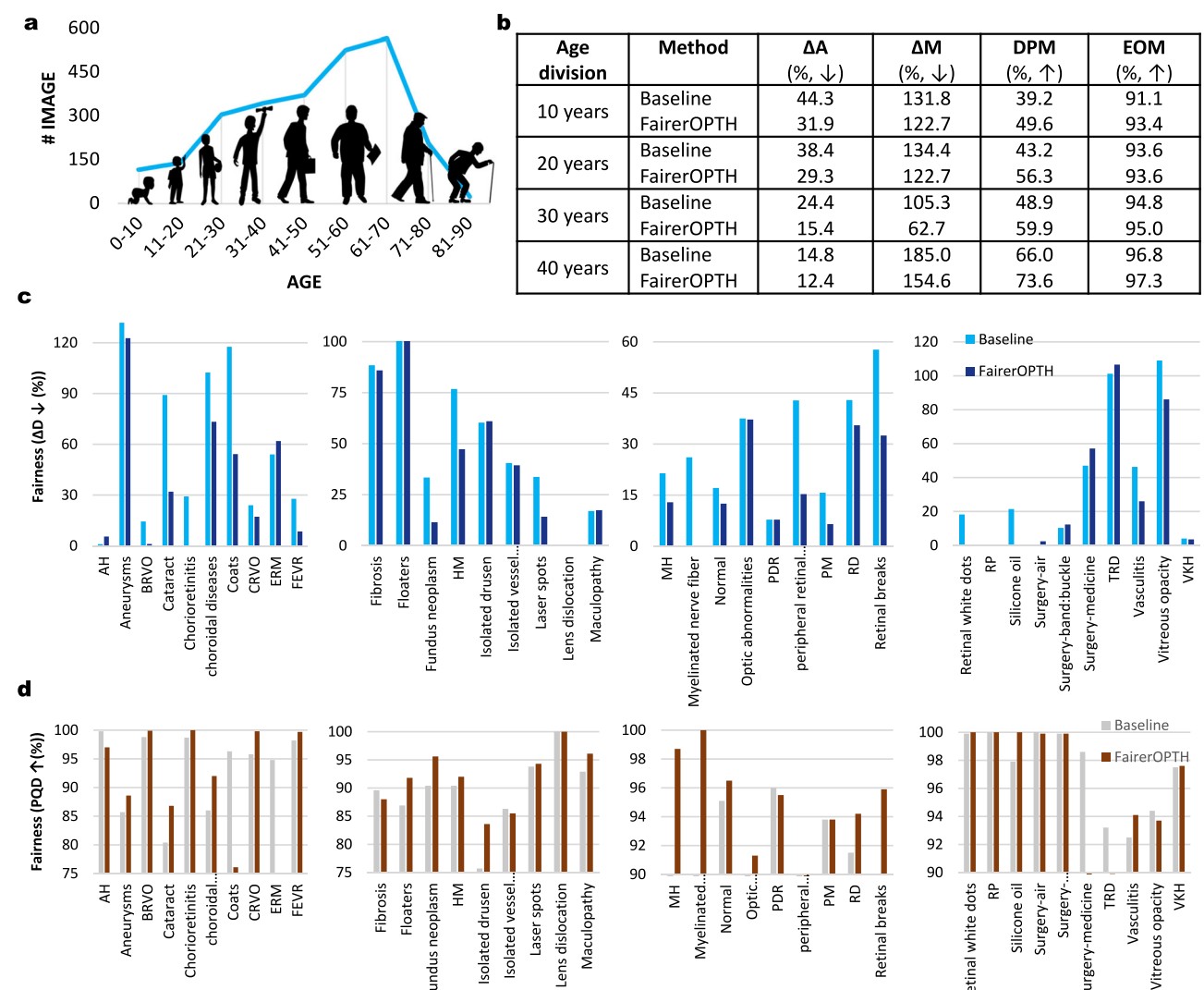

**Fig. 2 | Mitigating ageism of AI in ophthalmology. a** Sample distribution of patient age (in 10-year intervals) for the OculoScope test set. The unbalanced sample distribution of patient age may cause AI to favor the patients in the age group that contains more samples in the screening dataset, which will lead to the problem of AI ageism. In this study, we show that implicit fairness learning based on the relationship between ophthalmic diseases and fundus features can not only mitigate the unfairness that arises due to different sensitive attributes but also improve the screening accuracy for multiple diseases. **b** FairerOPTH is developed and validated with the OculoScope dataset, which contains data from 8405 patients with ages ranging from 0 to 90. Within each age division, FairerOPTH has obvious advantages in four fairness metrics ($\Delta A$ (average screening disparity), $\Delta M$ (max screening disparity), DPM (demographic disparity metric), and EOM (equality of opportunity metric)[2]) compared with the baseline model. The smaller $\Delta A$ and $\Delta M$ are, the better. **c** Details of the fairness metric $\Delta D$ (screening quality disparity) for 38 diseases. **d** Details of PQD (predictive quality disparity)[2] for 38 diseases. Source data are provided as a Source Data file.

demonstrate the effectiveness of using fundus features to enhance disease representation, thereby achieving higher diagnostic accuracy.

To further demonstrate the superiority of FairerOPTH, we also show the ROC curves and AUC values for each disease with the OculoScope dataset (Fig. 4c). Except for the AUCs of fibrosis, isolated vessel tortuosity, and optic abnormalities, which were lower than 0.95, the remaining 35 diseases all had AUCs exceeding 0.96. Among them, there are more than 20 disease types with AUC values above 0.99, showing that our method achieves good results in multilabel disease classification and can be adapted to most ophthalmic diseases. The false positive rate and false negative rate for typical diseases with imbalanced data are also demonstrated in Supplementary Table 2.

### Screening accuracy on the MixNAF dataset
In addition to validating the screening performance of FairerOPTH with UWF images, we also validated our method's performance with NAF images. To fully demonstrate the superiority of our method on the

MixNAF dataset, we also compare it with other state-of-the-art multi-label classification approaches. Each comparison method is slightly modified to identify the 16 types of disease included in the MixNAF dataset. The screening results are shown in Fig. 5 on the comparison with the narrow-angle dataset shows that our method exhibits the best mAP, sensitivity, and AUC. Specifically, the mAP is 2.2% higher than the second-best ASL method[18]. The results shows that our FairerOPTH also achieves good screening performance on narrow-angle fundus images partly from several public datasets. This screening improvement is mainly attributed to the introduction of fundus features.

### Generalizability of the FairerOPTH architecture
Other types of data such as commonly used text can be easily adapted to our architecture. To validate it, additional experiments using text as input to the pathology classification branch are conducted on the OculoScope dataset and denoted as "FairerOPTH (text)" (Supplementary Table 3). The input text is a description of fundus features in

**Table 1 | Demonstration of the mean average precision (mAP %) for each disease in each 10-year age bracket on the OculoScope dataset**

| Age condition | FairerOPTH/baseline (%) | | | | | | | | |
|---|---|---|---|---|---|---|---|---|---|
| Disease type | AH | Aneurysms | BRVO | Cataract | Chorioretinitis | Choroidal diseases | Coats | CRVO | ERM |
| 0–10 | 100/100 | 17.5/9.6 | - | - | - | 99.2/100 | 99.4/100 | - | - |
| 11–20 | - | 10/9.1 | - | 100/50 | - | 96.3/88.2 | 99/94.8 | 100/100 | - |
| 21–30 | - | 100/100 | - | 73.8/36.9 | 100/81.7 | 96.7/87.9 | 100/100 | 100/100 | 56.3/55.9 |
| 31–40 | 100/100 | 100/100 | - | 91.1/65.6 | 100/75 | 82.9/72.1 | 100/100 | 83.3/100 | 50.3/56.7 |
| 41–50 | - | 100/100 | 100/100 | 71.9/58.3 | 100/100 | 76.8/72.7 | 100/50 | 100/100 | 67.3/74.3 |
| 51–60 | 94.5/99.0 | 53.3/41.2 | 100/100 | 84.6/82.1 | - | 78.7/79.1 | 50.4/10.7 | 95.8/97.6 | 97.2/96.7 |
| 61–70 | 100/100 | 100/95.6 | 98.9/95.9 | 92.8/85.4 | - | 39.8/34.4 | - | 100/76.9 | 93/92.9 |
| 71–80 | 100/100 | 79.2/65.1 | 100/86.1 | 87.5/80.8 | - | 69.2/31 | - | 100/100 | 98.2/95.8 |
| 81–90 | - | 100/100 | 100/100 | 100/99 | - | 100/41.7 | - | - | 100/100 |

| Disease type | FEVR | Fibrosis | Floaters | Fundus neoplasm | HM | Isolated drusen | Isolated vessel tortuosity | Laser spots | Lens dislocation |
|---|---|---|---|---|---|---|---|---|---|
| 0–10 | 100/92.3 | 35.6/27.6 | - | 94.2/85.1 | - | - | 72.9/58.2 | 100/86.7 | - |
| 11–20 | 91.7/75.6 | 40.8/33.0 | 10/4.2 | 100/75.8 | 96.7/100 | 100/100 | 96.7/87.7 | 86.3/66.7 | - |
| 21–30 | 100/100 | 77.1/72.4 | 86.3/93.4 | 93/90.9 | 95.0/90.9 | 100/100 | 89.2/85.1 | 98.5/97.1 | - |
| 31–40 | 100/83.3 | 49.2/35.9 | 78.1/71 | 99.5/98.4 | 89.2/86.6 | 91.7/100 | 80.2/78.7 | 99.1/94.7 | 100/100 |
| 41–50 | - | 65.5/66.1 | 95.7/94.9 | 99/98 | 66.4/70.1 | 46/47 | 67.2/58.3 | 90.9/93.0 | 100/100 |
| 51–60 | - | 90.3/76.4 | 94.8/92.4 | 91/88.5 | 85.4/73.8 | 84.3/81.5 | 80.2/77.3 | 97.8/96.4 | 100/100 |
| 61–70 | - | 73.6/63.3 | 93.9/93.2 | 89/69.1 | 83.0/83.9 | 94.4/87.8 | 86.2/73.4 | 95.9/91.6 | 100/100 |
| 71–80 | - | 77.7/100 | 94.4/92.4 | 100/97.6 | 74.6/61.0 | 94.2/92.4 | 75.6/69.6 | 100/95.4 | 100/100 |
| 81–90 | - | - | 96.0/91.1 | - | 58.3/41.7 | 97/94.4 | 100/66.7 | - | - |

| Disease type | Maculopathy | MH | Myelinated nerve fiber | Normal | Optic abnormalities | PDR | Peripheral retinal degeneration | PM | RD | Vitreous opacity |
|---|---|---|---|---|---|---|---|---|---|---|
| 0–10 | - | 100/100 | 100/100 | 97.9/99.4 | 81.6/84.6 | 100/100 | 83.0/62.1 | - | 67.6/61.1 | - |
| 11–20 | 100/100 | - | - | 98.5/97.2 | 66.4/70.3 | - | 87.8/87.5 | 100/100 | 100/98.7 | 25/11.1 |
| 21–30 | 83.3/100 | 100/100 | - | 97.3/96.3 | 69.0/68.1 | 98.3/92.5 | 96.7/97.2 | 93.8/94.2 | 93.5/95.2 | 84.1/74.5 |
| 31–40 | 100/100 | 100/100 | - | 95.3/92.2 | 81.1/80.5 | 98.1/96.7 | 94.5/97.1 | 94.6/85.1 | 99.1/96.8 | 89.0/86.5 |
| 41–50 | 97.2/100 | 87.6/87.6 | 100/100 | 96.9/97.3 | 63.8/58.2 | 99.1/99.5 | 90.9/87.8 | 98.0/99.2 | 94.8/80.6 | 93.8/88.2 |
| 51–60 | 93.1/83.5 | 97.1/86.9 | 100/100 | 94.6/95.2 | 66.8/72.0 | 99.2/99.5 | 88.7/88.6 | 97.8/97.5 | 95.1/91.4 | 94.6/93.6 |
| 61–70 | 96.1/94.8 | 89.7/80.0 | 100/75.2 | 92.7/92.7 | 68.0/67.4 | 92.4/93.2 | 92.6/91.9 | 95.7/97.8 | 96.4/89.7 | 95.0/87.2 |
| 71–80 | 97.7/95.1 | 97.7/100 | 100/100 | 86.7/83.3 | 84.8/75.5 | 95.7/93.1 | 85.5/84.9 | 98.2/97.7 | 84.4/86.7 | 87.7/88.4 |
| 81–90 | 100/100 | - | - | - | 58.3/58.3 | - | 83.3/100 | 94.4/86.7 | - | - |

| Disease type | Retinal breaks | Retinal white dots | RP | Silicone oil | Surgery-air | Surgery-bandbuckle | Surgery-medicine | TRD | Vasculitis | VKH |
|---|---|---|---|---|---|---|---|---|---|---|
| 0–10 | - | - | 100/100 | 100/79.2 | 100/100 | - | 88.6/95.8 | 15.3/19.6 | 100/100 | - |
| 11–1–20 | - | - | 100/100 | 100/100 | - | - | 87.5/96.4 | - | 81/58.7 | 100/100 |
| 21–30 | 70.6/52.2 | 100/83.3 | 100/100 | 100/100 | 100/100 | - | - | 88.8/70.6 | 100/88.9 | 96.5/96.2 |
| 31–40 | 97.6/100 | 100/100 | 100/100 | 100/100 | 100/100 | 100/100 | - | 69.3/79.8 | 90.4/89.8 | 100/100 |
| 41–50 | 97.3/76.9 | - | 100/100 | 100/100 | 100/100 | 100/100 | 51.2/57.7 | 93.4/92.9 | 99.8/95.0 | 100/100 |
| 51–60 | 100/96.9 | 100/91.7 | 100/100 | 100/100 | 97.6/100 | 100/100 | 100/100 | 84.5/86.2 | 75.7/80.5 | 96.9/96.0 |
| 61–70 | 88.6/88.5 | - | 100/100 | 100/100 | 100/100 | 100/100 | 100/100 | 88.1/85.1 | 100/100 | 98.6/100 |
| 71–80 | 87.7/82.1 | - | 100/100 | 100/100 | 100/100 | 88.1/90 | - | - | 100/100 | - |
| 81–90 | - | - | - | - | - | - | - | - | - | - |

The mAP is the metric used for evaluating the accuracy of screening multiple diseases.

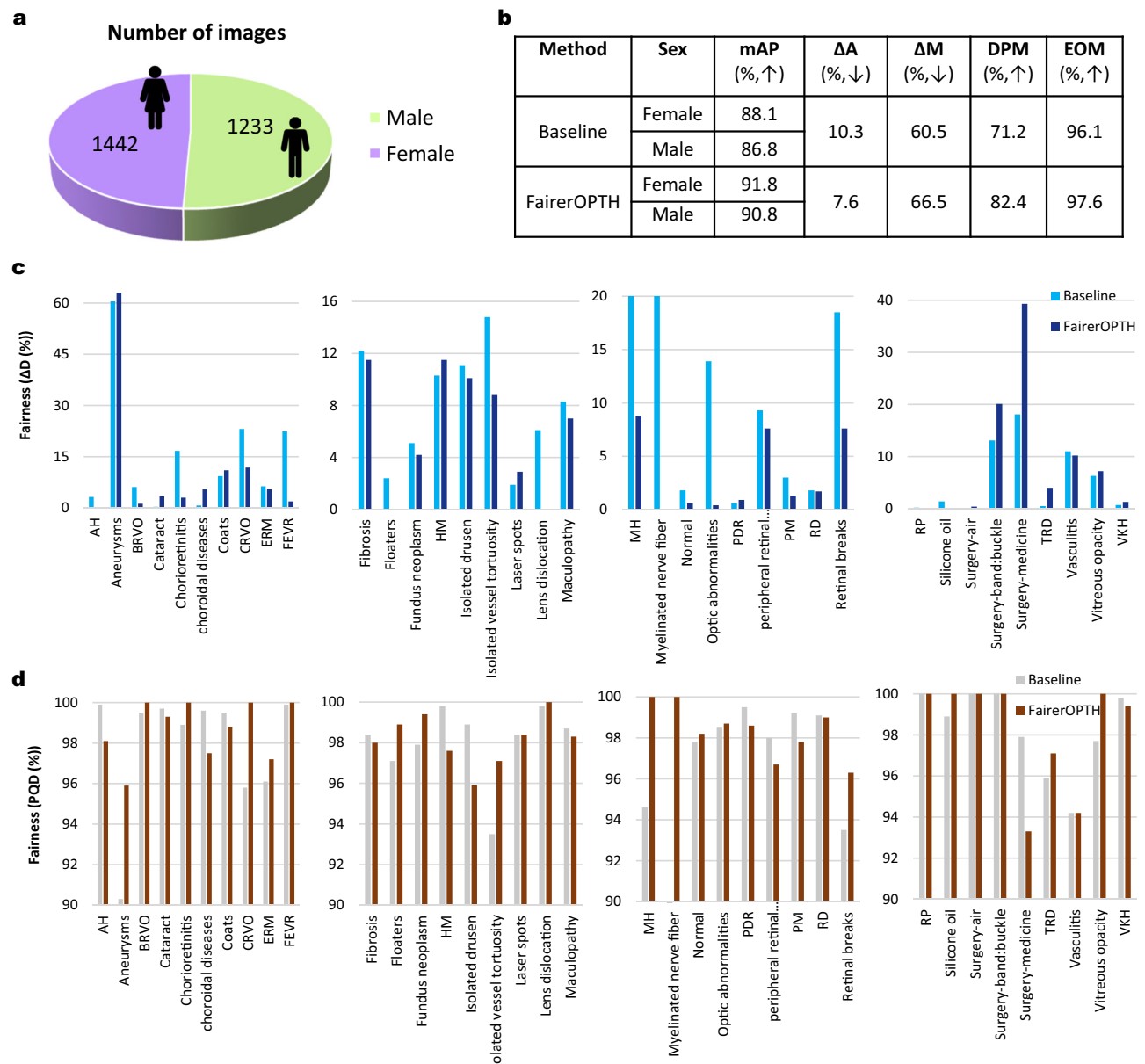

**Fig. 3 | Mitigating sexism of AI in ophthalmology. a** Proportion of male and female patients included in the OculoScope test set. **b** Comparison of fairness metrics (ΔA (average screening disparity), ΔM (max screening disparity), DPM (demographic disparity metric), and EOM (equality of opportunity metric)[2]) and screening accuracy (mAP) between the baseline and our FairerOPTH. **c** Details of the fairness metric ΔD (screening quality disparity) for 38 diseases. **d** Details of PQD (predictive quality disparity)[2] for 38 diseases. Source data are provided as a Source Data file.

the fundus image. We use the trained BERT[19] to extract features from the inputted text. The "FairerOPTH (text)" and "FairerOPTH (image)" achieve better performance than the Baseline. In addition, "FairerOPTH (text)" performs better in screening accuracy and fairness than "FairerOPTH (image)" most of the time. The results demonstrate that our architecture can adapt to the text modality and even achieve better performance.

The trained FairerOPTH model, like other common fundus image-based diagnostic models, can be directly deployed on fundus datasets without fundus feature annotations. As shown in Supplementary Fig. 1, the FairerOPTH model contains the upper and bottom two branches, where the upper branch is used to extract pathological features and help the bottom branch of disease classification. During the testing phase, similar to other fundus-based disease classification models, the trained FairerOPTH model only needs to input fundus images and can automatically extract pathological features and identify diseases. We

test the trained FairerOPTH model on the public IDRiD[20] dataset without fundus feature annotations. The diagnostic performance of the Baseline and ViT-Large[21] models is also shown in Supplementary Table 4.

**Ablation study**

To evaluate whether the newly annotated fundus features reflect an effective enhancement in disease diagnosis, we perform ablation studies by removing pathological information (*i.e.*, the pathology-aware attention module). Specifically, we use an advanced ASL[18] method ($L_{disea}$ term) to serve as the baseline model, which helps us to remove the potential confound of differing models. We also conduct an ablation study on the proposed pathological feature loss $L_{patho}$ to evaluate its effectiveness. The ablation study results on the OculoScope and MixNAF datasets are shown in Supplementary Table 5. The experiment demonstrates that the mAP for disease classification on

**Table 2 | Demonstration of the mAP for each disease in patients of different sexes on the OculoScope dataset**

FairerOPTH/Baseline (mAP, %)

| Sex condition | | | | | | | | | |
|---|---|---|---|---|---|---|---|---|---|
| Disease type | | AH | Aneurysms | BRVO | Cataract | Chorioretinitis | choroidal diseases | Coats | CRVO |
| Male | | 96.8/99.2 | 44.0/42.8 | 100/100 | 89.8/80.1 | 100/86.7 | 82.4/77.2 | 97.0/94.9 | 99.2/98.3 |
| Female | | 100/99.3 | 87.8/79.9 | 98.8/94.1 | 86.8/80.4 | 97/73.3 | 86.9/77.7 | 86.9/86.5 | 88.2/78.0 |

| Disease type | FEVR | Fibrosis | Floaters | Fundus neoplasm | HM | Isolated drusen | Isolated vessel tortuosity | Laser spots |
|---|---|---|---|---|---|---|---|---|
| Male | 98.1/94.4 | 75.4/64.0 | 93.2/92.8 | 96.2/92.4 | 89.6/74.5 | 84.3/80.1 | 82.2/76.3 | 94.3/92.5 |
| Female | 100/75.4 | 67.2/56.7 | 93.2/90.6 | 92.2/87.9 | 79.9/82.6 | 93.3/89.6 | 75.3/65.8 | 97.0/94.3 |

| Disease type | Maculopathy | MH | Myelinated nerve fiber | Normal | Optic abnormalities | PDR | peripheral retinal degeneration | PM |
|---|---|---|---|---|---|---|---|---|
| Male | 98.9/96.1 | 88.3/72.6 | 100/68.9 | 96.8/96.8 | 70.4/71.2 | 97.0/97.1 | 87.1/85.9 | 97.7/94.4 |
| Female | 92.2/88.4 | 96.4/95.5 | 100/98.9 | 96.2/95.0 | 70.7/62.0 | 97.9/97.7 | 93.9/94.3 | 96.4/97.3 |

| Disease type | Retinal white dots | RP | Silicone oil | Surgery-air | Surgery-band-buckle | Surgery-medicine | TRD | Vasculitis |
|---|---|---|---|---|---|---|---|---|
| Male | – | 100/100 | 99.9/98.6 | 100/100 | 81.7/88.7 | 86.9/90.0 | 80.0/81.3 | 90.3/88.0 |
| Female | 100/92.1 | 100/99.8 | 100/100 | 99.6/100 | 100/100 | 58.3/75 | 83.2/81.8 | 100/98.2 |

| Disease type | ERM | Vitreous opacity | Lens dislocation | VKH | RD | Retinal breaks |
|---|---|---|---|---|---|---|
| Male | 88.6/88.0 | 94.5/89.9 | 100/94.0 | 98.0/97.8 | 92.3/88.3 | 89.3/78.7 |
| Female | 93.6/93.7 | 87.9/84.4 | 100/100 | 99.2/98.5 | 93.9/89.8 | 96.4/94.7 |

the OculoScope dataset drops by 0.3% after removing the fundus features. Likewise, the mAP decreased by 2.0% after removing $L_{con}$ and $L_{joint}$ from the model, suggesting a strong relationship between fundus features and disease. Similar experimental results are shown for the MixNAF dataset. This performance gain once again demonstrates the effectiveness of our proposed method for enhancing disease diagnosis with fundus features.

## Discussion

Our extensive results demonstrate that the rich fundus features contained in UWF or NAF images can make AI models not only less unfair but also more accurate. We achieve this through the implicit fairness learning approach that incorporates the causal relationship between fundus features and eye diseases. This causal relationship is relatively independent of sensitive attributes and imaging modes. The fairness of our model is statistically significantly better than the baseline model (ResNet-101) using the disease classification branch alone. Even when screening many disease types with a significantly imbalanced distribution, experimental results demonstrate that FairerOPTH still greatly improves the accuracy and fairness of diagnosis for most diseases.

The discovery of using fundus features to mitigate the unfairness in the disease diagnosis is interesting because such an implicit learning method does not directly model fair regularization for AI diagnostic models and has never been reported. There are several common factors that cause biases[7,22]. For example, the biases have already been included in the collected dataset because of variations in incidence such as race, age, region, or sex. In addition, the biases may also be caused by missing data, such as missing samples for the target population, which prevents representation of the population. The widely used optimization objectives that are designed to minimize overall classification errors will benefit majority groups over minorities. To overcome these biases, pre-process[23–25], in-process[26,27], and post-process[28,29] have been demonstrated in recent literature. Different mechanism types have their own advantages and disadvantages. For instance, preprocessing mechanisms offer the advantage of compatibility with any classification algorithm, but using these methods may compromise the interpretability of the outcomes. Postprocessing mechanisms have the flexibility to be applied in conjunction with any classification algorithm. However, owing to their application at a relatively late stage in the learning process, they generally yield subpar outcomes. Our implicit learning method based on the relationship between the fundus features and disease diagnosis can be regarded as a kind of fair causal learning.

We also explore the most contributing factors by conducting experiments on the diseases including congenital retinal degenerative diseases (including retinitis pigmentosa, stargardt and other congenital macular diseases), optic abnormalities, and fibrosis that correspond to the three possible factors including congenital diseases with different manifestations in different periods, mixed causes, and great individual differences (Supplementary Table 6). The result demonstrates that compared with the baseline model, FairerOPTH has obvious improvement in fairness in Fibrosis disease, which indicates that individual difference is a potential factor that contributes most to mitigating unfairness. In addition, the mixed cause and congenital disease with different manifestations in different periods are also two relatively important factors.

FairerOPTH can aid in addressing the diagnostic basis by predicting the fundus features that exist in the input fundus images. This promotes transparency and explainability in fair models, which are vital for building trust in these models. Compared with regular screening methods, FairerOPTH can not only diagnose diseases in fundus images but also output the relevant fundus features at the same time. Because of their causal relationship, the fundus features can be used as the pathological explanation of the disease diagnosis on the

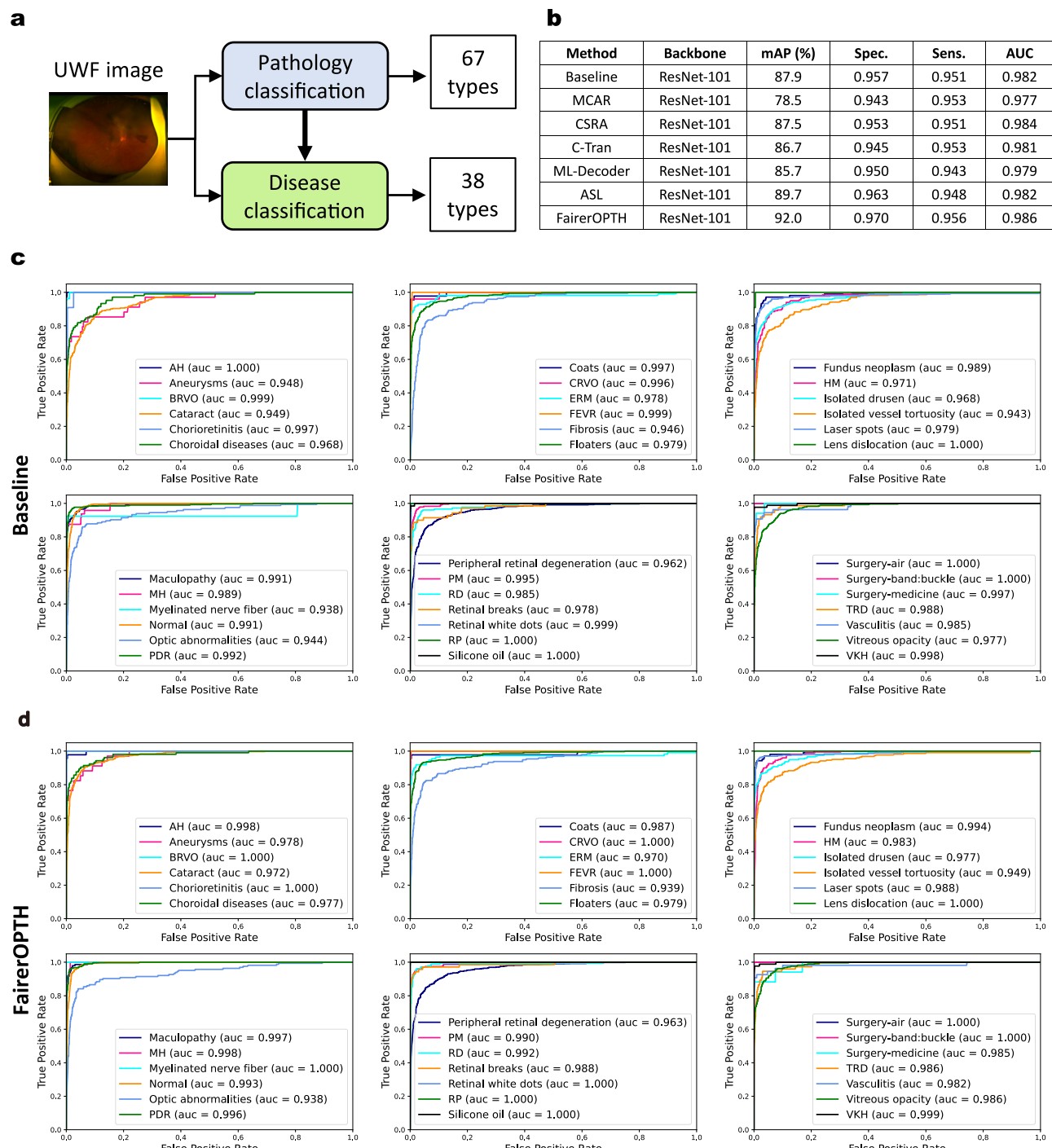

**Fig. 4 | Screening performance of the FairerOPTH on the OculoScope dataset.**
**a** The FairerOPTH consists of two branches, pathology and disease classification that predict 67 fundus features and 38 ophthalmic diseases, respectively. Such design of the two branches aims to enhance the disease representation in the disease classification branch, resulting in higher screening accuracy. **b** Comparison of FairerOPTH with the baseline model and state-of-the-art multi-label classification methods using mAP, specificity, sensitivity, and AUC (area under the curve) evaluation metrics. **c**, **d** ROC (Receiver Operating Characteristics) curves for 38 diseases of baseline model and FairerOPTH. Source data are provided as a Source Data file.

fundus image. This is different from traditional methods for disease explanation that uses the CAM[30] method to generate class activation maps as visual explanations (Supplementary Fig. 2). Such explanations are incomprehensible to non-professionals. In contrast, FairerOPTH can give the specific fundus feature name, which is beneficial to the patients with the disease in better understanding their clinical signs. However, there may be a contradiction between the predictions of the two branches, even the disease and pathological features are classified with high accuracy (Supplementary Table 7). This contradictory prediction does affect the accuracy of the model's interpretability, which should be addressed in future work.

Our approach has the potential to be extended to diagnosing other similar diseases. Apart from ophthalmic diseases, various other diseases have a diagnostic process that typically involves doctors identifying the pathological areas first and then considering all the pathological information to diagnose the specific disease in the

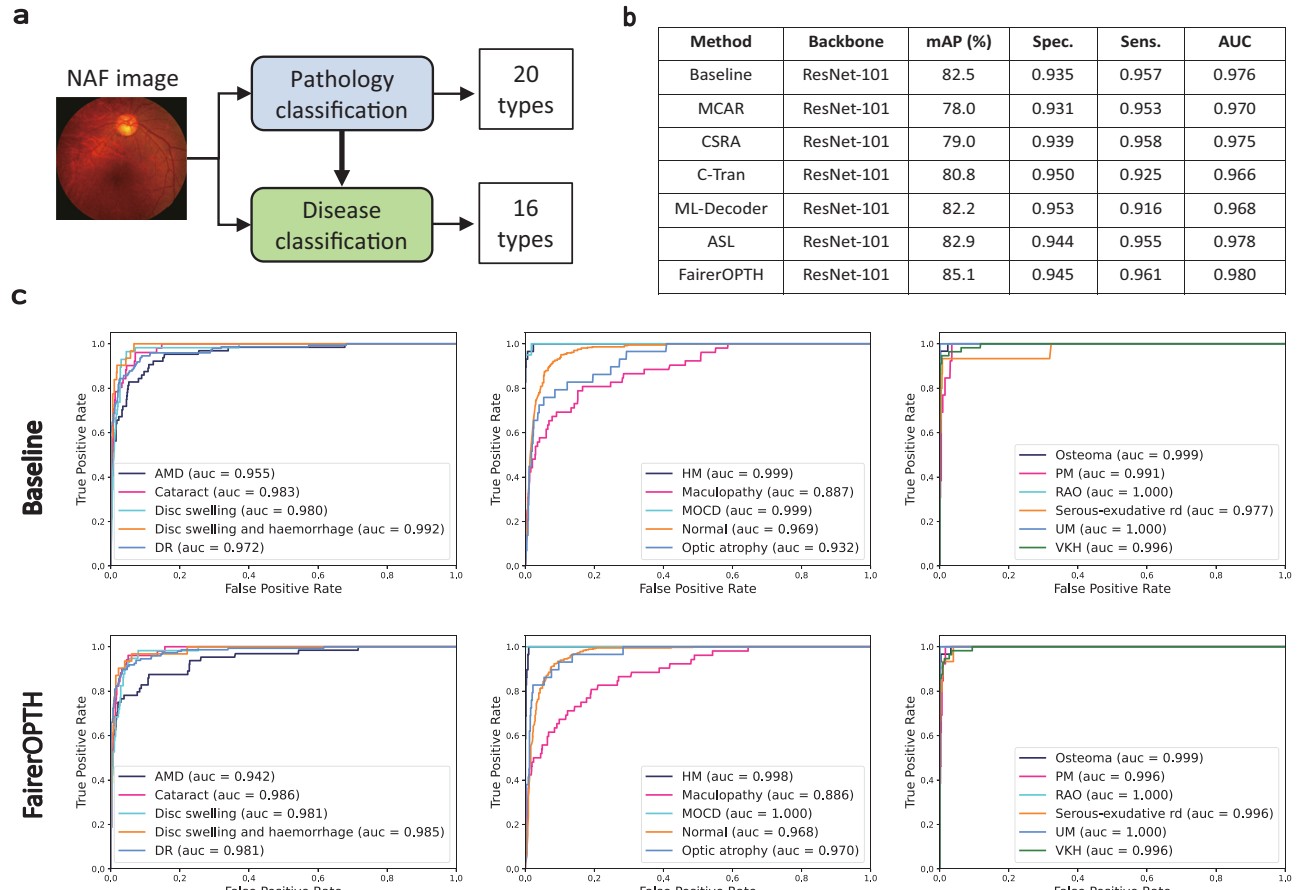

**Fig. 5 | Screening performance of the FairerOPTH on the MixNAF dataset. a** The FairerOPTH predicts 20 fundus features and 16 ophthalmic diseases from input NAF images. **b** Comparison of FairerOPTH with the baseline model and state-of-the-art multi-label classification methods using mAP, specificity, sensitivity, and AUC (area under the curve) evaluation metrics. **c** ROC (Receiver Operating Characteristics) curves for 16 diseases of baseline model and FairerOPTH. Source data are provided as a Source Data file.

patient. Therefore, our method can effectively be extended to similar diseases, thereby improving diagnosis and treatment outcomes.

Although FairerOPTH achieves good results, there are some limitations. First, there is a significant amount of fundus manifestation information in fundus images, with single images containing as many as six different pathologies. This pathological information is often scattered throughout the image. Therefore, accurately capturing these fundus features poses a significant challenge for the model and directly impacts the model's ability to effectively provide useful information for diagnosing diseases without introducing misinformation. In this study, we do not carefully design specific network modules to improve the identification accuracy of fundus features. However, the ablation study results demonstrate the significant role of fundus features. Therefore, designing a better deep network structure to improve the model's ability to capture diverse fundus features is a worthy pursuit for future research. Another challenge is the lack of pathological labels in most existing datasets, which primarily consists only of disease labels. This poses a substantial hurdle in studying the relationship between diseases and fundus manifestations. Using our labeled datasets, developing semi-supervised learning methods with larger public datasets is one of the ways to improve FairerOPTH.

Finally, we recognize the importance of considering and addressing potential disparities related to race and ethnicity in healthcare research. However, a limitation is that the demographics of our patient population can not perform the fairness evaluation to the sensitive attributes of race/ethnicity. Our hospital is situated in Shanghai, a region located in East China. The majority of our patients come from the surrounding provinces and the local area. In this geographical context, the Han nationality constitutes the overwhelming majority of our patient population. As a result, our dataset primarily comprises patients of Han ethnicity.

In summary, we have demonstrated that using the relationship between fundus features and disease diagnosis to model an implicit fairness learning approach can mitigate age and sex biases and improve the overall screening accuracy of the model. There are two main further directions of study. One is to develop other effective, generalizable implicit fairness learning approaches. The other is to evaluate whether our approach can be generalized to other diseases that are of worldwide concern to alleviate the unfairness brought about by AI.

## Methods

Ethical approval for the study was obtained from the Ethics Committee of Eye & ENT Hospital, Fudan University, Shanghai, China (Approval No. 2023427), and all procedures were in accordance with the Declaration of Helsinki. The study used anonymized images from patients who had completed their routine eye examinations or treatments at the hospital. Personal identifiers were removed post-annotation, ensuring that neither the retinal annotation experts nor the AI development team had access to patient-identifiable information. Given that the study utilized pre-existing, anonymized data and involved no additional examinations or interventions, the Ethics Committee waived the need for informed consent, citing minimal risk and adherence to privacy regulations.

## Datasets

We developed and validated a fairer AI system in ophthalmology (FairerOPTH) using a newly collected ultra-widefield fundus dataset (OculoScope) and a re-annotated narrow-angle dataset (MixNAF) combining multiple public datasets with data from over 8405 patients in total.

## OculoScope

The OculoScope dataset contains 16,530 UWF images captured by Optos P200dTx (Optos PLC, Dunfermline, United Kingdom) with an image resolution of 3070 × 3900 pixels from the Eye Institute and Department of Ophthalmology, Eye & ENT Hospital, Fudan University, Shanghai, China (Fig. 1). The Optos Daytona P200T utilizes advanced technology to acquire a singular 200° image that covers 82% of the retina within a duration of <0.4 seconds. The corresponding proprietary software, OptosAdvanceTM, facilitates documentation, monitoring, and analysis, including various measurement tools. Furthermore, multiple images can be merged to create a comprehensive 220° montage that covers 97% of the retina.

The images in the OculoScope dataset showed left and right eyes of 8405 patients of different sexes and ages, as well as fundus images of different disease stages and after treatment (Supplementary Table 8). The annotations were completed by two ophthalmologists with extensive clinical experience who strictly followed relevant standards and procedures during the data labeling process. Initially, the two ophthalmologists conducted two rounds of annotations on the same batch of fundus images, accounting for the symptom descriptions and final diagnosis results. This resulted in a total of four annotation rounds. Simultaneously, the fundus features closely associated with each disease type were annotated for each image. If there was a disagreement in the results, the two doctors would jointly arbitrate. In cases where a consensus could not be reached, a senior retinal disease expert would be consulted. The final annotation result was based on the consensus of these three retinal disease experts. The OculoScope dataset encompasses images depicting 38 disease types and 67 fundus features (Supplementary Data 1), such as retinitis pigmentosa (RP), pathological myopia (PM), familial exudative vitreoretinopathy (FEVR), central retinal vein occlusion (CRVO), von hippel-lindau disease (VHL), acute retinal necrosis (ARN), maculopathy, retinal detachment (RD), vogt-koyanagi-harada (VKH) disease, branch retinal vein Occlusion (BRVO), coats disease, lens dislocation, *etc*. Importantly, no other publicly available datasets include labels for fundus features (Supplementary Table 1). There are 2,423 and 512 normal fundus images in the training and test sets, respectively.

To demonstrate the complexity of OculoScope and the natural causal relationship that exists between disease and fundus features, we calculate the label densities for disease types and fundus features separately (Fig. 1e). The label density is denoted as $\rho$, with $\rho = \frac{1}{N}\sum_{i=1}^{C}|y_i|$, where $y_i$ is the number of images showing the $i$-th disease, $N$ represents the total number of images, and $C$ represents the number of disease types. The label densities for disease type and fundus features in OculoScope are 1.94 and 2.05, respectively. Such densities indicate that the fundus images in the OculoScope dataset contain more than one disease, while on average each fundus image nearly contains two fundus features. This also demonstrates that, on average, one disease corresponds to more than two fundus features.

## MixNAF

After thoroughly examining various publicly available fundus datasets, two expert doctors discovered a common limitation: while these datasets contained annotations for one or a few specific diseases, they did not cover a wide range of other eye diseases present in fundus images. To comprehensively evaluate the effectiveness of FairerOPTH, we took a multi-faceted approach. We combined images from partial datasets such as the JSIEC[4], Kaggle-EyePACS[31], iChallenge-AMD[32], OIA-ODIR[33], and RFMiD[34] datasets, along with newly collected regular narrow-angle fundus images captured using a traditional fundus camera. This amalgamation resulted in a new regular fundus dataset called MixNAF.

To ensure accurate annotations, we invited three ophthalmologists with extensive clinical experience to meticulously examine and determine the diagnosis for each of the collected images. Each image was carefully re-diagnosed and annotated with the corresponding fundus features. Images with unclear disease types and fundus features were excluded from the dataset. Ultimately, the MixNAF dataset consists of 4540 annotated fundus images, representing 16 diseases and 20 fundus features. There are 1622 and 365 normal fundus images in the training and test sets, respectively.

## FairerOPTH development

Based on the two fundus datasets captured by different imaging devices, we develop the FairerOPTH model (Supplementary Fig. 1) to fairly and accurately identify the diseases in fundus images simultaneously. Specifically, FairerOPTH takes a UWF image as an input and outputs the classification results showing any pathologies and the disease diagnosis. Our objective is to utilize the relationship between diseases and fundus features to learn informative and attribute-invariant representations to improve the screening accuracy of ophthalmic diseases and mitigate the sexism and ageism associated with AI screening.

FairerOPTH consists of two branches, the pathology classification and disease classification branches, involving two encoders and a pathology-aware attention module. Both encoders have an identical architecture: a pre-trained CNN (ResNet-101)[35]. The encoders are used to extract pathological features $F_p \in R^{C \times H \times W}$ and disease features $F_d \in R^{C \times H \times W}$, respectively. Then, the extracted pathological features are simultaneously input into a classifier and a pathology-aware attention module. Here, the pathology-aware attention module consists of a convolutional block attention module (CBAM)[36], which enables the computation of both channel and spatial attention. Finally, two classifiers are used for pathology and disease classification. We use three classification losses to constrain the classification results on pathology, disease, and consistency.

## Implicit fairness learning

Ophthalmologists rely on the observation of crucial pathological information to accurately diagnose ophthalmic diseases. By applying the pathology-aware attention module to analyze the extracted pathological features, we obtain a refined pathological feature map. Then, the refined feature map is added to the extracted disease features to obtain $F_e \in R^{C' \times H' \times W'}$, aiming to enhance disease features with pathological semantics. This module captures the inherent relationships between the pathological features and the disease categories, enhancing the representation of crucial information in the feature map. Specifically, the pathological attention module takes the pathological and disease features, namely $F_p$ and $F_d$, obtained by ResNet-101 as inputs and then learns the information for enhancing the disease features based on the pathological features.

Given the pathological feature map $F_p$, we first apply global average pooling to compress the spatial information, ultimately yielding global spatial features $F_{avg}$. Then, we input the global spatial features into a fully connected neural (FCN) layer to generate a channel attention map $W_{chn}$, which is described as follows:

$$W_{chn} = \sigma(ReLU(W_{FCN} \cdot F_{avg} + b_{FCN})) \tag{1}$$

where $\sigma(\cdot)$ is the sigmoid function normalizing the attention weights to [0,1], and $W_{FCN}$ and $b_{FCN}$ are the weights and bias of the FCN layer, respectively. Afterwards, we apply the Hadamard product to the learned attention weights $W_{chn}$ and the original feature map $F_p$ in order

to adaptively select the pathological information that is useful for disease classification.

$$F_p^{sel} = F_p \circ W_{chn} \qquad (2)$$

Finally, we add the selected feature map and the disease feature map $F_d$ in an element-wise manner to generate the pathologically aware feature map $F_p^{aware}$:

$$F_p^{aware} = F_p^{sel} \oplus F_d \qquad (3)$$

Therefore, the pathological attention module is used to capture the causal relationship between pathology and disease, and then further enhance the representation of diseases in the input fundus image. Finally, this module can make the model not only more accurate but also less biased.

## Loss functions

The distributions of diseases and pathologies are highly imbalanced distribution due to limited number of rare diseases, age groups, *etc.* in the dataset. To mitigate this problem, we use the asymmetric loss (ASL)[18] as the classification loss. The overall loss function is defined as

$$\mathcal{L}_{total} = \lambda_1 L_{patho} + \lambda_2 L_{disea} + \lambda_3 L_{joint} + \lambda_4 L_{consis} \qquad (4)$$

where $\lambda_1, \lambda_2, \lambda_3$, and $\lambda_4$ are the weights to balance the effects of different loss terms. $L_{patho}$ and $L_{disea}$ indicate the ASL loss for pathology and disease classification, respectively. Then, $L_{joint}$ is used to constrain the consistency of the two classification results, and $L_{consis}$ constrains the consistency of feature maps extracted from two encoders. Letting $F_p$ and $F_d$ be the two feature maps extracted from $Encoder_p$ and $Encoder_d$, respectively, $L_{consis}$ is defined as

$$\mathcal{L}_{consis} = \frac{1}{mnl} \sum_{i=0}^{m-1} \sum_{j=0}^{n-1} \sum_{k=0}^{l-1} (F_p(i,j,k) - F_d(i,j,k))^2 \qquad (5)$$

where $m$, $n$, and $l$ indicate the width, height, and channel of the features map, respectively.

## Implementation details

We use similar network configurations for the experiments with the OculoScope and MixNAF datasets. The parameters of the ResNet-101 network are initialized from the model pre-trained on the ImageNet[37]. Common data augmentation methods are used to enrich the data. For instance, we scale the size of input fundus images to 512 × 648 and augment the set of training data with random horizontal and vertical flips. The details of training and test dataset division are shown in Supplementary Table 9. For the OculoScope and MixNAF datasets, we carry out multi-class multi-disease classification and perform fairness analysis. The network is optimized using the Adam optimizer[38]. The initial learning rate is 1e-4, and a decaying learning rate is implemented by using OneCycleLR for each batch with a weight_decay=1e-4, betas=(0.9, 0.999)[39,40]. We trained the network for 100 epochs with a batch size of 16. The entire model is built on a Ubuntu 18.04 system with PyTorch and two NVIDIA GeForce RTX 2080Ti. The loss weight parameters of $\lambda_1$, $\lambda_2$, $\lambda_3$, and $\lambda_4$ are experimentally set to 0.01, 1, 0.01, and 1e-5, respectively.

## Fairness evaluation metrics

There exist several quantitative measures and approaches to ascertain fairness, encompassing widely employed metrics such as disparate impact and equalized odds. These fairness metrics aim to achieve complete decoupling between predictions and sensitive attributes, either unconditionally (disparate impact) or conditionally (equalized odds)[3]. We adopt three prediction-based fairness evaluation metrics: (i)

$\Delta D$ (screening quality disparity) measures the average precision (AP) difference between each sensitive group. We first calculate the difference between the highest and the lowest AP across different sensitive groups for a certain disease, and then compute the ratio between the obtained difference and the mean AP of all sensitive groups, i.e.,

$$\Delta D_k = SQD_k = \frac{\max_S(AP_{k,i}) - \min_S(AP_{k,i})}{\frac{1}{M}\sum_{i=1}^{M} AP_{k,i}} \qquad (6)$$

where $\mathbf{S} = \{1, 2, \cdots, M\}$ is the set of sensitive groups, for instance, under the sensitive attribute of sex, $\mathbf{S} = \{female, male\}$. $k$ is the index of a disease type, and $AP_{k,i}$ denotes the average precision of $i$-th group of a sensitive attribute in $k$ disease. Together, (ii) $\Delta M$ (max screening disparity) and (iii) $\Delta A$ (average screening disparity) are used to comprehensively evaluate the overall performance of the FairerOPTH. We calculate the $\Delta M$ and $\Delta A$ across sensitive groups for each category as follows:

$$\Delta M = \max_{k \in Q}(\Delta D_k) \qquad (7)$$

$$\Delta A = \frac{1}{K} \sum_{k=1}^{K} \Delta D_k \qquad (8)$$

where $\mathbf{Q} = \{1, 2, \cdots, K\}$ is the set of disease types. A model is considered to be fairer if it has smaller values for these three fair metrics. In addition to the above metrics, we also use three fairness evaluation metrics including PQD (predictive quality disparity), DPM (demographic disparity metric), and EOM (equality of opportunity metric), which is defined in the paper[2]. The PQD measures the fairness of an individual class, similar to our $\Delta D$ metric. The DPM and EOM are defined for multi-class multi-diseases, which are suitable for our experimental setting.

## Screening evaluation metrics

Each sample of the experimental datasets is assigned multiple labels of fundus features and disease types shown in the images. We treat the multi-label classification problem as multiple binary classification tasks involving the ≥16 disease classes and ≥20 fundus features. In this setting, the classification performance of each class is evaluated, which allows us to assess the classification performance of sensitive attributes. Specifically, $TP_j$, $FP_j$, $TN_j$, and $FN_j$ represent the true positive, false positive, true negative and false negative rates of the test samples for the $j$-th class. We report four evaluation metrics for each disease, including $Precision = \frac{TP_j}{FP_j + TP_j}$, $Specificity_j = \frac{TN_j}{FP_j + TN_j}$, $Sensitivity_j = \frac{TP_j}{TP_j + FN_j}$, and area under the ROC curve (AUC). Finally, we calculate the macro-average of the above metrics for each disease to measure the overall screening performance of a model.

## Models for comparison

The implementation of compared models such as https://github.com/gaobb/MCAR MCAR[14], https://github.com/Kevinz-code/CSRA CSRA[15], https://github.com/QData/C-Tran C-Tran[16], https://github.com/Alibaba-MIIL/ML_Decoder ML-Decoder[17], and https://github.com/Alibaba-MIIL/ASL ASL[18] adopts the official code and hyperparameter settings provided by the authors of the paper.

## Reporting summary

Further information on research design is available in the Nature Portfolio Reporting Summary linked to this article.

## Data availability

All data that support the findings of this study are included in the paper. This study utilizes some public datasets, including JSIEC (https://zenodo.org/record/3477553), Kaggle-EyePACS (https://www.kaggle.com/c/diabetic-retinopathy-detection), iChallenge-AMD

(https://ai.baidu.com/broad/download), OIA-ODIR (https://odir2019.grand-challenge.org/), and RFMiD (https://riadd.grand-challenge.org/download-all-classes/). The public IDRiD dataset can be accessed through the following link: https://ieee-dataport.org/open-access/indian-diabetic-retinopathy-image-dataset-idrid. Our newly collected OculoScope and reorganized MixNAF datasets are publicly available at figshare https://figshare.com/s/926c2c2ef9e77ab5eb9d. Source data are provided with this paper.

## Code availability

The codes and trained models associated with FairerOPTH are freely available at https://github.com/mintanwei/Fairer-AI(https://doi.org/10.5281/zenodo.10892893), which is based on PyTorch.

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

## Acknowledgements

We gratefully acknowledge support for this work provided by National Natural Science Foundation of China (NSFC) (Grant No.: 82020108006

to C.Z., U2001209 to B.Y., and 62372117 to W.T.), and Natural Science Foundation of Shanghai (NSFS) (Grant No.: 21ZR1406600 to W.T.).

## Author contributions

W.T., Q.W., B.Y., and C.Z. conceived the framework and technique for eye disease research. B.Y., Z.X., W.T., and H.F. designed and implemented the fairer AI algorithm. B.Y., W.T., Q.W., and H.F. designed the validation experiments. C.Z., Q.W., H.K. H.F., and Y.L. collected, organized, and annotated the UWF and NAF fundus images. H.F. trained the network and performed the validation experiments. W.T., Z.X., and Q.W. analyzed the validation results. C.Z., B.Y., and Y.L. supervised the research. All authors had access to the study data and contributed to writing the paper.

## Competing interests

The authors declare no competing interests.
