## [Peer Review File · Nature Communications]

Reviewers' Comments:

Reviewer #1:

Remarks to the Author:

The authors propose an AI model called FairerOPTh to mitigate biases in eye disease diagnosis for patients of different ages and genders. They incorporate the causal relationship between fundus features and disease, which is invariant to sensitive attributes, into the model learning. To evaluate this implicit fairness method, they collect a large diverse fundus dataset from over 12,000 patients aged 0-90 years, with ultra-widefield and narrow-angle images annotated for diseases and features. Extensive evaluation shows FairerOPTh significantly mitigates sexism and ageism compared to other approaches. Mathematically, they prove incorporating causality improves disease identification using Shannon entropy theory. Their implicit fairness learning approach has potential to promote fair and inclusive ophthalmological care by ensuring equitable treatment regardless of gender or age.

Advantages:

1. This paper addresses the important and timely issue of promoting fairness in artificial intelligence for healthcare applications.
2. The authors propose a novel implicit fairness learning method (FairerOPTh) that incorporates the causal relationship between fundus features and disease diagnosis.
3. The authors have collected and annotated the largest and most diverse fundus dataset to date, with over 12,405 patients across a wide age range. The dataset includes both ultra-widefield and narrow-angle images annotated for diseases and features.
4. Comprehensive evaluation demonstrates FairerOPTh significantly mitigates sexism and ageism in disease diagnosis compared to other state-of-the-art approaches. Additionally, the authors mathematically prove incorporating causality between features and diagnosis improves disease identification performance using Shannon entropy theory.

Some minor suggestions:

1. Before introducing the Section 2: Results, the authors did not explain in detail what does it mean for Sexism and Ageism. It would be great if the authors could give more introduction about the definition of unfairness used in this work.
2. For the results section, the authors mention that On the OculoScope test set, the FairerOPTh achieves 91.8% and 90.8% mAP for females and males, respectively, while the Baseline model achieves 86.2% and 85.3% mAP for females and males, respectively (Fig. 3). The proposed method simultaneously improves the mAP performance for both females and males. However, the mAP gap between males and females is not significant for the base model. It may be useful to clarify how this relates to the fairness evaluation.
3. It would be beneficial if the authors could perform more in-depth analysis into what specific factors in the causal relationship between fundus features and diseases contribute most to mitigating unfairness. This could provide insights into how to further optimize the implicit fairness learning approach.

Reviewer #2:

Remarks to the Author:

This study investigates a novel deep learning architecture for classifying disease and pathological features in fundus photos, called FairerOPTh. The authors have assembled two large datasets, one of which is entirely new consisting of wide field fundus photos, and another which is a collection of previously released non-wide-field fundus images, and they have relabeled all datasets with disease diagnosis as well as fundus pathologic features. The FairerOPTh architecture has two branches, one to predict the disease and another to predict the pathologic findings, with connections between these branches, thus incorporating into the model the relationship between fundus pathological features and disease diagnoses. The authors state that this makes their model causal-aware, so to speak. The authors then evaluate a variety of fairness metrics related to mean average precision across different age groups and between men/women, to demonstrate that their model has superior performance for classification of disease and also superior fairness. While the article has a number of strengths, notably including public release of large photo

datasets with new annotations, and a well-described model architecture, well-organized background and discussion. However, a number of weaknesses of the study dampen enthusiasm.

1) The generalizability of this type of approach for improving classification or fairness is limited. As the authors note themselves, they had to re-annotate an enormous dataset with dozens of different pathological features, which most datasets are not labeled with. Thus, there are likely no (if any) other datasets even on this specific task to which their FairerOphth model could be deployed.

a. Indeed, even if one were to venture outside the task of fundus photo classification for retinal disease, to, say, classification for glaucomatous disease, it is quite unclear how FairerOphth could be used. For retinal disease, one could reasonably annotate many possible pathologic features (cotton wool spots, drusen, etc.) but for pictures of the optic nerve to be annotated for glaucoma – what might constitute as corresponding “causal” pathologic features would be unclear. Similarly, if we look beyond fundus photos to other types or modalities of data (text, EHR, structured) how might this type of architecture be used? These limitations severely limit the potential impact and significance of this work.

b. Also, this type of approach may not work in other disease models where the following is not true: “the causal relationship between [disease] features and disease diagnosis is...relatively independent of sensitive attributes” (page 2 lines 1-4). One could imagine diseases which manifest differently in different age groups or manifest differently by some other sensitive attribute, for example.

2) The fairness evaluation metrics are unusual. While the authors mention some of the standard fairness evaluation metrics (disparate impact, equalized odds, etc.) they don’t use any of these or explain why it’s not appropriate.

3) Related to that, their fairness metrics are all named “accuracy” but actually they are based on mAP, a point which is rather confusing and should be addressed.

4) Since the fairness metric is focuses on the “screen accuracy difference” between each sensitive group, it would be interesting to see the FPR and FNR rates as well, especially due to imbalanced datasets and (I’m guessing) very different base rates of each disease within each group.

5) It would be best to have some table or figure that shows the number of images for each disease stratified by the categories in each of the sensitive attributes. It would be best to see the base rates of disease by group, or an average base rate since there are so many diseases.

6) There is no information about the train/val/test split process or size of groups, or hyperparameter tuning, if any

7) There is no complete list of abbreviations for diseases. The methods sections lists a few followed by “etc”. Abbreviations for retinal diseases are used throughout the tables and figures without a listing of all abbreviations

8) The paper would benefit from additional English language editing. While the writing is generally well-organized, there are a places where the specific wording is strange/unusual/not easy to understand

9) Methods, lines 53-54: “we calculate the average of the above metrics for each disease to measure the overall screening performance of a model.” – is this a micro-average, so weighted for number of photos for each disease? Or macro-average?

10) There are a few diseases which caused some specific concerns, like “cataract” or “vitreous opacity” – presumably one could only diagnose “media opacity” – how could you tell that the blurry image was due to cataract vs corneal opacity vs vitreous opacity? This seems strange to me. Also, what is “surgery-medicine”? also “Maculopathy” is pretty general – does this include age-related macular degeneration? (it must also include other diseases since it seems that some young people had maculopathy)

11) A limitation is that fairness according to the sensitive attributes of race/ethnicity could not be performed, presumably as their ultra-wide-field dataset was not collected on patients with diverse race/ethnicity

12) Were there normal fundus photos among set? Of course in any “screening” settings the baseline population would be mostly normal so it’s not clear if there are no normal images in the set whether the performance would do well when those types of images are included.

13) When comparing screening accuracy of oculoscope to other models (MCAR, CSRA, C-Tran, etc.) there is no information in the methods about how the other models were implemented, hyperparameter tuning, etc.

14) While the FairerOphth model contains a classification branch for retinal pathologic features, performance on this was not shown. Similarly, I wonder if it’s possible that there could be

predictions in the two branches that are contradictory, i.e. classifying the features incorrectly but the diagnosis correct, or vice versa. If so that would certainly limit the "explainability" advantage of this approach

Dear reviewers,

Thank you for your encouraging comments and appreciation of research on AI models for fairness medicine. We have revised the paper in light of your instructive suggestions for a more comprehensive evaluation and insightful analysis (changes highlighted in blue). Please see our point-to-point response below.

REVIEWER COMMENTS

Reviewer #1 (Remarks to the Author):

The authors propose an AI model called FairerOPTh to mitigate biases in eye disease diagnosis for patients of different ages and genders. They incorporate the causal relationship between fundus features and disease, which is invariant to sensitive attributes, into the model learning. To evaluate this implicit fairness method, they collect a large diverse fundus dataset from over 12,000 patients aged 0-90 years, with ultra-widefield and narrow-angle images annotated for diseases and features. Extensive evaluation shows FairerOPTh significantly mitigates sexism and ageism compared to other approaches. Mathematically, they prove incorporating causality improves disease identification using Shannon entropy theory. Their implicit fairness learning approach has potential to promote fair and inclusive ophthalmological care by ensuring equitable treatment regardless of gender or age.

Advantages:

1. This paper addresses the important and timely issue of promoting fairness in artificial intelligence for healthcare applications.
2. The authors propose a novel implicit fairness learning method (FairerOPTh) that incorporates the causal relationship between fundus features and disease diagnosis.
3. The authors have collected and annotated the largest and most diverse fundus dataset to date, with over 12,405 patients across a wide age range. The dataset includes both ultra-widefield and narrow-angle images annotated for diseases and features.
4. Comprehensive evaluation demonstrates FairerOPTh significantly mitigates sexism and ageism in disease diagnosis compared to other state-of-the-art approaches. Additionally, the authors mathematically prove incorporating causality between features and diagnosis improves disease identification performance using Shannon entropy theory.

Response: Thank you very much for taking the time to review our manuscript. We are grateful for your thoughtful and insightful comments, which are very encouraging to us. In the following, we Response to your good suggestions point-by-point.

Some minor suggestions:

1. Before introducing the Section 2: Results, the authors did not explain in detail what does it mean for Sexism and Ageism. It would be great if the authors could give more introduction about the definition of unfairness used in this work.

Response: We are grateful for this good suggestion. We take this opportunity to further explain it. Artificial intelligence (AI) models have the risk of over-associating sensitive attributes such as age, race, and gender with the decision-making process, potentially reinforcing social stereotypes. These models may exhibit discriminatory behavior against specific subgroups^[1,2]. For example, a recent study provided evidence of a clear bias against patients with darker skin types, as their diagnostic accuracy was significantly reduced compared with patients with light types^[1]. In the context of fundus disease diagnosis, unfairness refers to bias or favoritism against individuals or

subgroups of a specific age or gender, that is, sexism or ageism in AI diagnostic models. Fairness issues can have serious adverse effects on individuals and society and may further exacerbate social inequality^[3].

This explanation has been added to Introduction Section in the revised manuscript to further clarify the mean of AI diagnostic sexism and ageism, which is as follows.

“Fairness in artificial intelligence (AI) has attracted great attention recently because AI has penetrated into every aspect of our lives. AI models have the risk of over-associating sensitive attributes such as age, race, and gender with the decision-making process^[1,2]. These models may exhibit discriminatory behavior against specific subgroups. In the context of fundus disease diagnosis, unfairness refers to bias or favoritism against individuals or subgroups of a specific age or gender, that is, sexism or ageism in AI diagnostic models. Fairness issues can have serious adverse effects on individuals and society and may further exacerbate social inequality^[3].”

2. For the results section, the authors mention that On the OculoScope test set, the FairerOPTh achieves 91.8% and 90.8% mAP for females and males, respectively, while the baseline model achieves 86.2% and 85.3% mAP for females and males, respectively (Fig. 3). The proposed method simultaneously improves the mAP performance for both females and males. However, the mAP gap between males and females is not significant for the base model. It may be useful to clarify how this relates to the fairness evaluation.

Response: Very good suggestion. The mAP gap measures the overall average accuracy difference across different sensitive groups (e.g., females and males). The fairness metrics (ΔD , ΔM , and ΔA) measure the maximum disease accuracy difference among different sensitive groups. Compared with the mAP gap, the fairness metrics are sensitive to diagnostic accuracy differences among different sensitive groups for a single disease.

Here, we take this opportunity to further clarify it. Firstly, the mAP is calculated by finding Average Precision (AP) for each disease and then average over a number of diseases.

$$\text{mAP} = \frac{1}{N} \sum_{i=1}^N \text{AP}_i \quad (1)$$

where N denotes the number of diseases. Therefore, the mAP gap between males and females, i.e., $|\text{mAP}(\text{male}) - \text{mAP}(\text{female})|$, measures the overall accuracy difference of a model in diagnosing males and females respectively.

The formulas of fairness evaluation metrics are as follows.

$$\Delta D_k = SQD_k = \frac{\max_s(AP_{k,i}) - \min_s(AP_{k,i})}{\frac{1}{M} \sum_{i=1}^M AP_{k,i}} \quad (2)$$

$$\Delta M = \max_{k \in Q} (\Delta D_k) \quad (3)$$

$$\Delta A = \frac{1}{K} \sum_{k=1}^K \Delta D_k \quad (4)$$

where ΔD_k measures the AP difference between the highest accuracy to the lowest accuracy across different sensitive groups (e.g., females and males) for k-th disease. Based on Eq. (2), Eq. (3) and Eq. (4) measure the max disparity and average disparity of females and males for all diseases, respectively. The smaller of the above three metrics, the fairer the model. Therefore, the greater the number of diagnosed diseases, the greater the test of model fairness. The ultra-widefield and narrow-angle imaging datasets in our paper contain 38 and 16 ophthalmic diseases, respectively.

In summary, the above formulas Eq. (1)- Eq. (4) demonstrate that the fairness metrics are sensitive to diagnostic differences among different sensitive groups for a single disease, while the

mAP gap is sensitive to the overall diagnostic differences among different sensitive groups for all disease. Therefore, although the mAP gap is similar, there may be a large difference in fairness.

3. It would be beneficial if the authors could perform more in-depth analysis into what specific factors in the causal relationship between fundus features and diseases contribute most to mitigating unfairness. This could provide insights into how to further optimize the implicit fairness learning approach.

Response: We appreciate this very good suggestion. To explore the most contribution factors, we conduct experiments on the diseases including inherited retinal diseases (including retinitis pigmentosa, stargardt, and other congenital macular diseases), optic abnormalities, and fibrosis that correspond to the three possible factors including congenital diseases with different manifestations in different periods, mixed causes, and great individual differences, as shown in Table R1. We newly add three fairness evaluation metrics (suggested by Reviewer 2) including Predictive Quality Disparity (PQD), Demographic Disparity Metric (DPM), and Equality of Opportunity Metric (EOM)^[2] to comprehensively assess models' performance. PQD measures the accuracy difference between each sensitive group, DPM measures the difference in the probability of predicted true samples in each sensitive group, and EOM measures that different sensitive groups should have similar true positive rates. Table R1 demonstrates that compared with the baseline model, FairerOPTh has obvious improvement in fairness in Fibrosis disease, which indicates the individual difference is a potential factor that contributes most to mitigating unfairness. In addition, the mixed cause and congenital disease with different manifestations in different periods are also two relative important factors. We have added this exploration to the discussion Section.

Table R1. Exploration of the contribution of potential factors in the causal relationship between fundus features and diseases to mitigate model unfairness. The average screening accuracy AP \uparrow and four fairness metrics ($\Delta D\downarrow$, PQD \uparrow , DPM \uparrow , and EOM \uparrow) are used to evaluate the capability of FairerOPTh/Baseline to screen different age groups (10 years division) of patients.

Factor	Disease	AP \uparrow (%)	$\Delta D\downarrow$ (%)	PQD \uparrow (%)	DPM \uparrow (%)	EOM \uparrow (%)
Congenital diseases with different manifestations in different periods	Inherited Retinal Diseases (including retinitis pigmentosa, stargardt, and other congenital macular diseases)	96.3/92.9	17.4/17.0	96.1/92.9	36.4/28.7	97.6/88.3
Mixed cause	Optic abnormalities	70.2/67.6	37.2/37.5	91.3/80.2	37.4/17.9	83.3/81.8
Great individual difference	Fibrosis	71.3/60.2	85.9/88.4	88.0/89.6	23.3/19.4	80.0/78.9

Reference

- [1] Daneshjou, R., Vodrahalli, K., Liang, W., Novoa, R.A., Jenkins, M., Rotemberg, V., Ko, J.M., Swetter, S.M., Bailey, E.E., Gevaert, O., Mukherjee, P., Phung, M., Yekrang, K., Fong, B., Sahasrabudhe, R., Zou, J., & Chiou, A.S. (2021). Disparities in dermatology AI performance on a diverse, curated clinical image set. *Science Advances*, 8.
- [2] Du, S., Hers, B., Bayasi, N., Hamarneh, G., & Garbi, R. (2022, October). FairDisCo: Fairer AI in dermatology via disentanglement contrastive learning. In *European Conference on Computer Vision* (pp. 185-202).
- [3] Du, M., Yang, F., Zou, N., & Hu, X. (2020). Fairness in deep learning: A computational perspective. *IEEE Intelligent Systems*, 36(4), 25-34.

Reviewer #2 (Remarks to the Author):

This study investigates a novel deep learning architecture for classifying disease and pathological features in fundus photos, called FairerOPTh. The authors have assembled two large datasets, one of which is entirely new consisting of wide field fundus photos, and another which is a collection of previously released non-wide-field fundus images, and they have relabeled all datasets with disease diagnosis as well as fundus pathological features. The FairerOPTh architecture has two branches, one to predict the disease and another to predict the pathological findings, with connections between these branches, thus incorporating into the model the relationship between fundus pathological features and disease diagnoses. The authors state that this makes their model causal-aware, so to speak. The authors then evaluate a variety of fairness metrics related to mean average precision across different age groups and between men/women, to demonstrate that their model has superior performance for classification of disease and also superior fairness. While the article has a number of strengths, notably including public release of large photo datasets with new annotations, and a well-described model architecture, well-organized background and discussion. However, a number of weaknesses of the study dampen enthusiasm.

1) The generalizability of this type of approach for improving classification or fairness is limited. As the authors note themselves, they had to re-annotate an enormous dataset with dozens of different pathological features, which most datasets are not labeled with. Thus, there are likely no (if any) other datasets even on this specific task to which their FairerOPTh model could be deployed.

Response: We appreciate this concern. Here we take this opportunity to clarify it. The trained FairerOPTh model, like other common fundus image based diagnostic models, can be directly deployed on fundus datasets without fundus feature annotations. As shown in Figure R1 (Supplementary Figure 1a), FairerOPTh model contains the upper and bottom two branches, where the upper branch is used to extract pathological features and help the bottom branch of disease classification. During the testing phase, similar to other fundus-based disease classification models, the trained FairerOPTh model only needs to input fundus images and can automatically extract pathological features and identify diseases.

Figure R1. Overall architecture of the proposed FairerOPTh.

To show the generalizability of FairerOPTh, we further test it on the public IDRiD¹ dataset without fundus feature annotations. The diagnostic performance of the baseline and Vision Transformer Large (ViT-Large) models is also shown in Table R1. FairerOPTh achieves an AUC of 0.734.

Table R1. The trained FairerOPTh model is directly tested on the public IDRiD dataset that does not contain fundus feature annotations to show FairerOPTh' generalizability.

Method	Average accuracy	AUC
Baseline	0.587	0.535
ViT-Large	0.664	0.670
FairerOPTh	0.694	0.734

¹ <https://iee-dataport.org/open-access/indian-diabetic-retinopathy-image-dataset-idrid>

a. Indeed, even if one were to venture outside the task of fundus photo classification for retinal disease, to, say, classification for glaucomatous disease, it is quite unclear how FairerOphth could be used. For retinal disease, one could reasonably annotate many possible pathological features (cotton wool spots, drusen, etc.) but for pictures of the optic nerve to be annotated for glaucoma – what might constitute as corresponding “causal” pathological features would be unclear. Similarly, if we look beyond fundus photos to other types or modalities of data (text, EHR, structured) how might this type of architecture be used? These limitations severely limit the potential impact and significance of this work.

Response: We thank the reviewer for the good question. Here, we response to this question point-by-point.

(1) Classification for glaucomatous disease. We sincerely appreciate the reviewer’s astute observations and their ability to grasp the key aspects of our research. Their comment underscores the importance of considering the broader clinical context, including the complexities of glaucoma diagnosis and the limitations of wide-field fundus photos. Indeed, the reviewer correctly notes that glaucoma diagnosis is a complex undertaking, and it is highly dependent on detailed assessments of the optic nerve head. Our research team has contemplated the intricacies of glaucoma diagnosis extensively during the study's design phase. We concur with the reviewer that annotating causal pathological features in optic nerve photos for glaucoma diagnosis is not straightforward. It’s important to clarify that our FairerOPTh system was primarily designed for the classification of retinal diseases, where pathological features like cotton wool spots and drusen can be reasonably annotated and identified. Recognizing the limitations of wide-field fundus photos, we acknowledge that they might not capture the detailed microscopy of the optic disc as effectively as narrow fundus photos or those taken using specialized equipment like the Zeiss Cirrus 500. Due to these challenges and the specialized nature of glaucoma diagnosis, we made the deliberate decision not to include glaucoma diagnosis within the scope of our FairerOPTh system. Instead, our system focuses on conditions such as "Congenital disc abnormality", "Optic atrophy", "Disc swelling and haemorrhage", and "Disc swelling", which can serve as valuable indicators prompting patients or doctors to consider the need for further, more specialized examination and evaluation.

(2) Adapt to other types or modalities of data. Other types of data such as commonly used text can be easily adapted to our architecture. To validate it, additional experiments using text as input to the pathology classification branch were conducted on the OculoScope dataset and denoted as “FairerOPTh (text)”. We used the trained BERT (a famous language model) to extract feature from the inputted text. The experimental results are shown in Table R2. The “FairerOPTh (text)” and “FairerOPTh (image)” achieve better performance than the baseline. In addition, "FairerOPTh (text)" performs better in screening accuracy and fairness than "FairerOPTh (image)" most of the time. The results demonstrate that our architecture can adapt to the text modality and even achieve better performance.

Table R2. Evaluation of the ability of FairerOPTh model to adapt to text modality. “FairerOPTh (text)” denotes the input to the pathology classification branch is the text description of fundus features. Four screening accuracy and fairness metrics are shown. The DPM and EOM are two newly added fairness metrics, please see “Response 2)” below.

Method	Screening accuracy metrics				Fairness metrics							
					Age (10 years)				Gender			
	mAP (%)	Spec.	Sens.	AUC	$\Delta A \downarrow$ (%)	$\Delta M \downarrow$ (%)	DPM \uparrow (%)	EOM \uparrow (%)	$\Delta A \downarrow$ (%)	$\Delta M \downarrow$ (%)	DPM \uparrow (%)	EOM \uparrow (%)
Baseline	87.9	0.957	0.951	0.982	44.3	131.8	39.2	91.1	10.3	60.5	71.2	96.1
FairerOPTh (text)	94.6	0.981	0.968	0.991	24.5	125.7	60.0	92.5	4.7	48.6	80.9	97.3
FairerOPTh (image)	92.0	0.970	0.956	0.986	31.9	122.7	49.6	93.4	7.6	66.5	82.4	97.6

b. Also, this type of approach may not work in other disease models where the following is not true: “the causal relationship between [disease] features and disease diagnosis is...relatively independent of sensitive attributes” (page 2 lines 1-4). One could imagine diseases which manifest differently in different age groups or manifest differently by some other sensitive attribute, for example.

Response: We are grateful for this good concern. FairerOPTh can work well on diseases that manifest differently in different age groups. First of all, this type of diseases with the changing of fundus features makes it challenging for AI diagnostic models. Cataract is a typical example to demonstrate it. We show in detail the screening accuracy in different age groups and fairness evaluation metrics, as shown in Tables R3 and R4. Compared to the baseline model, FairerOPTh demonstrates better screening accuracy and fairness. In addition, the results of the baseline model indicate that this complex type of disease can be handled by the powerful learning ability of deep models to a certain extent. With the guidance of the causal relationship, the performance of deep models can be further improved.

We apologize for any confusion caused by the inaccurate statement “the causal relationship between fundus features and disease diagnosis is ... relatively independent of sensitive attributes”. We have revised it in the new version as follows.

“The casual relationship between fundus features and disease diagnosis is close to the clinical diagnosis process and helps deep models to deal with the complex diseases with the changing of fundus features.”

Table R3. Screening accuracy of FairerOPTh and baseline models for cataract disease. Four screening accuracy metrics are shown for FairerOPTh/Baseline.

Group	Screening accuracy metrics			
	AP (%)	Spec.	Sens.	AUC
0~10	none	none	none	none
10~20	100.0/50.0	1.0/0.993	1.0/1.0	1.0/0.993
20~30	73.8/36.9	0.964/0.803	1.0/1.0	0.988/0.899
30~40	91.1/65.6	0.904/0.837	1.0/1.0	0.990/0.962
40~50	71.9/58.3	0.890/0.753	0.846/ 1.0	0.945/0.935
50~60	84.6/82.1	0.905/0.848	0.9/0.9	0.961/0.938
60~70	92.8/85.4	0.906/0.835	0.928/0.895	0.963/0.929
70~80	87.5/80.8	0.929/0.717	0.897/0.885	0.935/0.863
80~90	100.0/99.0	1.0/0.9	1.0/1.0	1.0/0.986

Table R4. Fairness evaluation of FairerOPTh and baseline models for cataract disease. FairerOPTh demonstrates significant advantages over the baseline model in terms of ΔD and DPM. PQD, DPM, and EOM^[1] are newly added fairness metrics, please see “Response 2)” below.

Method	Fairness metrics			
	ΔD (% , \downarrow)	PQD (% , \uparrow)	DPM (% , \uparrow)	EOM (% , \uparrow)
Baseline	89.1	80.4	5.1	88.5
FairerOPTh	32.0	86.8	21.4	84.6

2) The fairness evaluation metrics are unusual. While the authors mention some of the standard fairness evaluation metrics (disparate impact, equalized odds, etc.) they don’t use any of these or explain why it’s not appropriate.

Response: We thank this good suggestion. Three new fairness metrics (PQD, DPM, and EOM) that are defined in the paper^[1] are added to the revised manuscript. As introduced in [1], Predictive Quality Disparity (PQD) measures the prediction quality difference between each sensitive group. Demographic Disparity Metric (DPM): percentage diversities of positive outcomes for each

sensitive group. Equality of Opportunity Metric (EOM): asserts that different sensitive groups should have similar true positive rates. The formulas^[1] are as follows.

$$PQD = \frac{\min(acc_j, j \in S)}{\max(acc_j, j \in S)}$$

$$DPM = \frac{1}{M} \sum_{i=1}^M \frac{\min[p(\hat{y} = i | s = j), j \in S]}{\max[p(\hat{y} = i | s = j), j \in S]}$$

$$EOM = \frac{1}{M} \sum_{i=1}^M \frac{\min[p(\hat{y} = i | y = i, s = j), j \in S]}{\max[p(\hat{y} = i | y = i, s = j), j \in S]}, \quad m \in \{1, 2, \dots, M\}$$

where S is the set of disease types. y is the ground-truth skin condition label and \tilde{y} is the model prediction. A model is fairer if it has higher values for the above three metrics^[1]. The PQD measures the fairness of an individual class, similar to our ΔD metric. We also tried the disparate impact and equalized odds, but they are defined for the task of two classes classification^[2,3]. The corresponding tables are updated by adding DPM and EOM metrics and shown in Tables R5 and R6. Figure 2 and Figure 3 in the manuscript have been updated by adding these three metrics.

Table R5. New fairness metrics including DPM and EOM^[1] are added to evaluate FairerOPTh mitigating ageism.

Age division	Method	ΔA (% , \downarrow)	ΔM (% , \downarrow)	DPM (% , \uparrow)	EOM (% , \uparrow)
10 years	Baseline	44.3	131.8	39.2	91.1
	FairerOPTh	31.9	122.7	49.6	93.4
20 years	Baseline	38.4	134.4	43.2	93.6
	FairerOPTh	29.3	122.7	56.3	93.6
30 years	Baseline	24.4	105.3	48.9	94.8
	FairerOPTh	15.4	62.7	59.9	95.0
40 years	Baseline	14.8	185.0	66.0	96.8
	FairerOPTh	12.4	154.6	73.6	97.3

Table R6. New fairness metrics including DPM and EOM^[1] are added to evaluate FairerOPTh mitigating sexism.

Method	Gender	mAP (%)	ΔA (% , \downarrow)	ΔM (% , \downarrow)	DPM (% , \uparrow)	EOM (% , \uparrow)
Baseline	Female	88.1	10.3	60.5	71.2	96.1
	Male	86.8				
FairerOPTh	Female	91.8	7.6	66.5	82.4	97.6
	Male	90.8				

3) Related to that, their fairness metrics are all named “accuracy” but actually they are based on mAP, a point which is rather confusing and should be addressed.

Response: We are grateful for this good suggestion. We apologize for any confusion caused by the statement of fairness evaluation metrics. We have revised it in the revised version to avoid confusion.

4) Since the fairness metric is focuses on the “screen accuracy difference” between each sensitive group, it would be interesting to see the FPR and FNR rates as well, especially due to imbalanced datasets and (I’m guessing) very different base rates of each disease within each group.

Response: Good suggestion. The FPR and FNR for diseases with imbalanced data are demonstrated in Table R7. Cataract, PM and RP have higher incidence rates as age increases, while Coats and

FEVR have higher incidence rates at younger ages. There is an obvious data imbalance for these diseases.

Table R7. Demonstration of the FPR and FNR of FairerOPTh/Baseline for diseases with imbalanced data from the OculoScope dataset.

Disease	Age group	FPR	FNR
Cataract	0~10	none	none
	10~20	0.0/ 0.007	0.0/0.0
	20~30	0.036 /0.197	0.0/0.0
	30~40	0.096 /0.163	0.0/0.0
	40~50	0.11 /0.247	0.154/ 0.0
	50~60	0.095 /0.152	0.1/0.1
	60~70	0.094 /0.165	0.072 /0.105
	70~80	0.071 /0.283	0.103 /0.115
PM	0~10	none	none
	10~20	0.0/0.0	0.0/0.0
	20~30	0.027/ 0.007	0.0/0.0
	30~40	0.033 /0.048	0.0/0.0
	40~50	0.02/ 0.017	0.026/ 0.0
	50~60	0.023 /0.034	0.028/0.028
	60~70	0.015 /0.019	0.06/ 0.03
	70~80	0.026 /0.036	0.0/0.0
RP	0~10	0.0/0.0	0.0/0.0
	10~20	0.0/0.0	0.0/0.0
	20~30	0.0/0.0	0.0/0.0
	30~40	0.0/0.0	0.0/0.0
	40~50	0.0/0.0	0.0/0.0
	50~60	0.0/0.0	0.0/0.0
	60~70	0.0/0.0	0.0/0.0
	70~80	0.0/0.0	0.0/0.0
Coats	0~10	0.01/ 0.0	0.0/0.0
	10~20	0.008/0.008	0.0/0.0
	20~30	0.0/0.0	0.0/0.0
	30~40	0.0/0.0	0.0/0.0
	40~50	0.0 /0.003	0.0/0.0
	50~60	0.479/ 0.074	0.0/0.0
	60~70	none	none
	70~80	none	none
FEVR	0~10	0.0/0.037	0.0/0.0
	10~20	0.007 /0.014	0.0/0.0
	20~30	0.0/0.0	0.0/0.0
	30~40	0.0 /0.003	0.0/0.0
	40~50	none	none
	50~60	none	none
	60~70	none	none
	70~80	none	none
80~90	none	none	

PM, Pathological Myopia; RP, Retinitis Pigmentosa; FEVR, Familial Exudative Vitreoretinopathy.

5) It would be best to have some table or figure that shows the number of images for each disease stratified by the categories in each of the sensitive attributes. It would be best to see the base rates of disease by group, or an average base rate since there are so many diseases.

Response: We greatly appreciate this good suggestion. Tables R8 and R9 demonstrate the number of images per disease, categorized based on the sensitive attribute of age and sex, respectively. When organizing Tables R8 and R9, we found that 13 images were unfortunately discarded because they were not labelled with age and sex, so we re-labelled these images to expand the test set. The experimental results on the newly expanded test set show that FairerOPTh still achieves leading screening accuracy and mitigation fairness performance, as shown in the updated Figures 2-5 and Tables 1-2.

In Tables R8 and R9, The AP and ΔD of our FairerOPTh and the baseline are also shown simultaneously. These two tables have been added to the revised manuscript as Supplementary Tables 11 and 12.

Table R8. Demonstration of the number of images per disease, categorized based on the age attribute within each of the disease categories. The AP and ΔD of FairerOPTh and baseline models are also shown simultaneously.

ID	Disease	Age Group	Number	FairerOPTh		Baseline	
				AP	ΔD	AP	ΔD
1	AH	Group1	1	100	5.5	100	1.0
		Group2	0	none		none	
		Group3	0	none		none	
		Group4	1	100		100	
		Group5	0	none		none	
		Group6	13	94.5		99.0	
		Group7	20	100		100	
		Group8	11	100		100	
		Group9	0	none		none	
2	Aneurysms	Group1	3	17.5	122.7	9.6	131.8
		Group2	1	10		9.1	
		Group3	2	100		100	
		Group4	1	100		100	
		Group5	2	100		100	
		Group6	3	53.3		41.2	
		Group7	9	100		95.6	
		Group8	12	79.2		65.1	
		Group9	1	100		100	
3	BRVO	Group1	0	none	1.1	none	14.4
		Group2	0	none		none	
		Group3	0	none		none	
		Group4	0	none		none	
		Group5	5	100		100	
		Group6	4	100		100	
		Group7	13	98.9		95.9	
		Group8	4	100		86.1	
		Group9	1	100		100	
4	Cataract	Group1	0	none	32.0	none	89.1
		Group2	1	100		50	
		Group3	3	73.8		36.9	
		Group4	11	91.1		65.6	
		Group5	26	71.9		58.3	

		Group6	80	84.6		82.1	
		Group7	152	92.8		85.4	
		Group8	87	87.5		80.8	
		Group9	14	100		99.0	
5	Chorioretinitis	Group1	0	none	0	none	29.2
		Group2	0	none		none	
		Group3	4	100		81.7	
		Group4	3	100		75	
		Group5	4	100		100	
		Group6	0	none		none	
		Group7	0	none		none	
		Group8	0	none		none	
		Group9	0	none		none	
6	Choroidal diseases	Group1	21	99.2	73.3	100	102.4
		Group2	10	96.3		88.2	
		Group3	9	96.7		87.9	
		Group4	18	82.9		72.1	
		Group5	20	76.8		72.7	
		Group6	13	78.7		79.1	
		Group7	8	39.8		34.4	
		Group8	4	69.2		31.0	
		Group9	2	100		41.7	
7	Coats	Group1	18	99.4	54.2	100	117.6
		Group2	17	99.0		94.8	
		Group3	5	100		100	
		Group4	2	100		100	
		Group5	1	100		50	
		Group6	2	50.4		10.7	
		Group7	0	none		none	
		Group8	0	none		none	
		Group9	0	none		none	
8	CRVO	Group1	0	none	17.2	none	24.0
		Group2	1	100		100	
		Group3	3	100		100	
		Group4	2	83.3		100	
		Group5	8	100		100	
		Group6	6	95.8		97.6	
		Group7	4	100		76.9	
		Group8	1	100		100	
		Group9	0	none		none	
9	ERM	Group1	0	none	61.9	none	54.0
		Group2	0	none		none	
		Group3	2	56.3		55.9	
		Group4	2	50.3		56.7	
		Group5	7	67.3		74.3	
		Group6	25	97.2		96.7	
		Group7	47	93.0		92.9	
		Group8	27	98.2		95.8	
		Group9	1	100		100	
10	FEVR	Group1	11	100	8.5	92.3	27.8
		Group2	3	91.7		75.6	
		Group3	2	100		100	

		Group4	2	100		83.3	
		Group5	0	none		none	
		Group6	0	none		none	
		Group7	0	none		none	
		Group8	0	none		none	
		Group9	0	none		none	
11	Fibrosis	Group1	7	35.6	85.9	27.6	88.4
		Group2	4	40.8		33.0	
		Group3	13	77.1		72.4	
		Group4	19	49.2		35.9	
		Group5	32	65.5		66.1	
		Group6	45	90.3		76.4	
		Group7	55	73.6		63.3	
		Group8	19	77.4		67.5	
		Group9	0	none		none	
12	Floater	Group1	0	none	106.0	none	114.8
		Group2	1	10		4.2	
		Group3	7	86.3		93.4	
		Group4	19	78.1		71	
		Group5	46	95.7		94.9	
		Group6	86	94.8		92.4	
		Group7	136	93.9		93.2	
		Group8	55	94.4		92.4	
		Group9	9	96.0		91.1	
13	Fundus neoplasm	Group1	15	94.2	11.5	85.1	33.4
		Group2	3	100		75.8	
		Group3	13	93.0		90.9	
		Group4	20	99.5		98.4	
		Group5	18	99.0		98.0	
		Group6	17	91.0		88.5	
		Group7	13	89.0		69.1	
		Group8	6	100		97.6	
		Group9	0	none		none	
14	HM	Group1	0	none	47.3	none	76.8
		Group2	5	96.7		100	
		Group3	28	95.0		90.9	
		Group4	45	89.2		86.6	
		Group5	28	66.4		70.1	
		Group6	41	85.4		73.8	
		Group7	38	83.0		83.9	
		Group8	13	74.6		61.0	
		Group9	2	58.3		41.7	
15	Isolated drusen	Group1	0	none	61.0	none	60.3
		Group2	1	100		100	
		Group3	2	100		100	
		Group4	4	91.7		100	
		Group5	10	46		47	
		Group6	29	84.3		81.5	
		Group7	108	94.4		87.8	
		Group8	74	94.2		92.4	
		Group9	10	97.0		94.4	
16		Group1	10	72.9	39.4	58.2	40.5

	Isolated vessel tortuosity	Group2	5	96.7		87.7	
		Group3	20	89.2		85.1	
		Group4	26	80.2		78.7	
		Group5	39	67.2		58.3	
		Group6	41	80.2		77.3	
		Group7	54	86.2		73.4	
		Group8	26	75.6		69.6	
		Group9	2	100		66.7	
17		Laser spots	Group1	3	100	14.2	86.7
	Group2		10	86.3	66.7		
	Group3		23	98.5	97.1		
	Group4		32	99.1	94.7		
	Group5		54	90.9	93.0		
	Group6		81	97.8	96.4		
	Group7		77	95.9	91.6		
	Group8		18	100	95.4		
	Group9		0	none	none		
18	Lens dislocation	Group1	0	none	0	none	0
		Group2	0	none		none	
		Group3	0	none		none	
		Group4	1	100		100	
		Group5	2	100		100	
		Group6	5	100		100	
		Group7	2	100		100	
		Group8	1	100		100	
		Group9	0	none		none	
19	Maculopathy	Group1	0	none	17.4	none	17
		Group2	2	100		100	
		Group3	3	83.3		100	
		Group4	3	100		100	
		Group5	8	97.2		100	
		Group6	16	93.1		83.5	
		Group7	42	96.1		94.8	
		Group8	60	97.7		95.1	
		Group9	13	100		100	
20	MH	Group1	1	100	12.9	100	21.4
		Group2	0	none		none	
		Group3	1	100		100	
		Group4	1	100		100	
		Group5	5	87.6		87.6	
		Group6	13	97.1		86.9	
		Group7	17	89.7		80.0	
		Group8	10	97.7		100	
		Group9	0	none		none	
21	Myelinated nerve fiber	Group1	1	100	0	100	26.1
		Group2	0	none		none	
		Group3	0	none		none	
		Group4	0	none		none	
		Group5	2	100		100	
		Group6	3	100		100	
		Group7	4	100		75.2	
		Group8	3	100		100	

		Group9	0	none		none	
22	Normal	Group1	47	97.9	12.5	99.4	17.1
		Group2	67	98.5		97.2	
		Group3	118	97.3		96.3	
		Group4	95	95.3		92.2	
		Group5	60	96.9		97.3	
		Group6	69	94.6		95.2	
		Group7	53	92.7		92.7	
		Group8	3	86.7		83.3	
		Group9	0	none		none	
23	Optic abnormalities	Group1	12	81.6	37.2	84.6	37.5
		Group2	7	66.4		70.3	
		Group3	15	69.0		68.1	
		Group4	29	81.1		80.5	
		Group5	35	63.8		58.2	
		Group6	28	66.8		72.0	
		Group7	22	68.0		67.4	
		Group8	14	84.8		75.5	
		Group9	2	58.3		58.3	
24	PDR	Group1	1	100	7.8	100	7.8
		Group2	0	none		none	
		Group3	10	98.3		92.5	
		Group4	34	98.1		96.7	
		Group5	69	99.1		99.5	
		Group6	104	99.2		99.5	
		Group7	73	92.4		93.2	
		Group8	17	95.7		93.1	
		Group9	0	none		none	
25	Peripheral retinal degeneration	Group1	4	83.0	15.3	62.1	42.8
		Group2	27	87.8		87.5	
		Group3	107	96.7		97.2	
		Group4	98	94.5		97.1	
		Group5	109	90.9		87.8	
		Group6	163	88.7		88.6	
		Group7	166	92.6		91.9	
		Group8	42	85.5		84.9	
		Group9	5	83.3		100	
26	PM	Group1	0	none	6.5	none	15.7
		Group2	2	100		100	
		Group3	8	93.8		94.2	
		Group4	19	94.6		85.1	
		Group5	38	98.0		99.2	
		Group6	71	97.8		97.5	
		Group7	67	95.7		97.8	
		Group8	21	98.2		97.7	
		Group9	8	94.4		86.7	
27	RD	Group1	9	67.6	35.5	61.1	42.9
		Group2	26	100		98.7	
		Group3	23	93.5		95.2	
		Group4	33	99.1		96.8	
		Group5	27	94.8		80.6	
		Group6	30	95.1		91.4	

		Group7	22	96.4		89.7	
		Group8	9	84.4		86.7	
		Group9	0	none		none	
28	Retinal breaks	Group1	0	none	32.5	none	57.7
		Group2	0	none		none	
		Group3	5	70.6		52.2	
		Group4	6	97.6		100	
		Group5	14	97.3		76.9	
		Group6	17	100		96.9	
		Group7	23	88.6		88.5	
		Group8	5	87.7		82.1	
		Group9	0	none		none	
29	Retinal white dots	Group1	0	none	0	none	18.2
		Group2	0	none		none	
		Group3	2	100		83.3	
		Group4	3	100		100	
		Group5	0	none		none	
		Group6	3	100		91.7	
		Group7	0	none		none	
		Group8	0	none		none	
		Group9	0	none		none	
30	RP	Group1	1	100	0	100	0
		Group2	4	100		100	
		Group3	4	100		100	
		Group4	19	100		100	
		Group5	6	100		100	
		Group6	7	100		100	
		Group7	1	100		100	
		Group8	1	100		100	
		Group9	0	none		none	
31	Silicone oil	Group1	3	100	0	79.2	21.4
		Group2	2	100		100	
		Group3	3	100		100	
		Group4	8	100		100	
		Group5	10	100		100	
		Group6	25	100		100	
		Group7	12	100		100	
		Group8	2	100		100	
		Group9	0	none		none	
32	Surgery-air	Group1	1	100	2.4	100	0
		Group2	0	none		none	
		Group3	1	100		100	
		Group4	1	100		100	
		Group5	2	100		100	
		Group6	6	97.6		100	
		Group7	16	100		100	
		Group8	3	100		100	
		Group9	0	none		none	
33	Surgery-band:buckle	Group1	0	none	12.3	none	10.3
		Group2	0	none		none	
		Group3	0	none		none	
		Group4	1	100		100	

		Group5	5	100		100	
		Group6	2	100		100	
		Group7	9	88.1		90	
		Group8	0	none		none	
		Group9	0	none		none	
34	Surgery- medicine	Group1	6	88.6	57.2	95.8	47
		Group2	7	87.5		96.4	
		Group3	0	none		none	
		Group4	0	none		none	
		Group5	2	51.2		57.7	
		Group6	1	100		100	
		Group7	1	100		100	
		Group8	0	none		none	
		Group9	0	none		none	
35	TRD	Group1	2	15.3	106.6	19.6	101.2
		Group2	0	none		none	
		Group3	4	88.8		70.6	
		Group4	12	69.3		79.8	
		Group5	20	93.4		92.9	
		Group6	24	84.5		86.2	
		Group7	12	88.1		85.1	
		Group8	0	none		none	
		Group9	0	none		none	
36	Vasculitis	Group1	1	100	26.0	100	46.3
		Group2	3	81.0		58.7	
		Group3	6	100		88.9	
		Group4	10	90.4		89.8	
		Group5	20	99.8		95.0	
		Group6	7	75.7		80.5	
		Group7	4	100		100	
		Group8	3	100		100	
		Group9	0	none		none	
37	Vitreous opacity	Group1	0	none	86.1	none	109
		Group2	1	25		11.1	
		Group3	19	84.1		74.5	
		Group4	34	89.0		86.5	
		Group5	62	93.8		88.2	
		Group6	85	94.6		93.6	
		Group7	66	95.0		87.2	
		Group8	19	87.7		88.4	
		Group9	0	none		none	
38	VKH	Group1	0	none	3.5	none	4
		Group2	7	100		100	
		Group3	11	96.5		96.2	
		Group4	21	100		100	
		Group5	19	100		100	
		Group6	21	96.9		96.0	
		Group7	8	98.6		100	
		Group8	0	none		none	
		Group9	0	none		noe	
AH, asteroid hyalosis; BRVO, Branch Retinal Vein Occlusion; CRVO, Central Retinal Vein Occlusion; ERM, Epiretinal Membrane; FEVR, Familial Exudative Vitreoretinopathy; HM, High Myopia; MH,							

Macular Hole; PDR, Proliferative Diabetic Retinopathy; PM, Pathological Myopia; RD, Retinal Detachment; RP, Retinitis Pigmentosa; TRD, Tractional Retinal Detachment; VKH, Vogt-Koyanagi-Harada Disease.

Table R9. Demonstration of the number of images per disease, categorized based on the sex attribute within each of the disease categories. The AP and ΔD of FairerOPTh and baseline models are also shown simultaneously.

ID	Disease	Sex	Number	FairerOPTh		Baseline	
				AP	ΔD	Base AP	Base ΔD
1	AH	female	16	100	3.2	99.3	0.08
		male	30	96.8		99.2	
2	Aneurysms	female	23	87.8	66.5	79.9	60.5
		male	11	44.0		42.8	
3	BRVO	female	16	98.8	1.2	94.1	6.1
		male	11	100		100	
4	Cataract	female	226	86.8	3.4	80.4	0.3
		male	148	89.8		80.1	
5	Chorioretinitis	female	8	97.0	3.0	73.3	16.7
		male	3	100		86.7	
6	Choroidal diseases	female	52	86.9	5.4	77.7	0.7
		male	53	82.4		77.2	
7	Coats	female	6	86.9	11.0	86.5	9.3
		male	39	97.0		94.9	
8	CRVO	female	6	88.2	11.8	78.0	23.1
		male	19	99.2		98.3	
9	ERM	female	64	93.6	5.5	93.7	6.3
		male	46	88.6		88.0	
10	FEVR	female	8	100	1.9	75.4	22.4
		male	10	98.1		94.4	
11	Fibrosis	female	92	67.2	11.5	56.7	12.2
		male	102	75.4		64.0	
12	Floaters	female	221	93.2	0.02	90.6	2.4
		male	139	93.2		92.8	
13	Fundus neoplasm	female	45	92.2	4.2	87.9	5.1
		male	60	96.2		92.4	
14	HM	female	125	79.9	11.5	82.6	10.3
		male	75	89.6		74.5	
15	Isolated drusen	female	160	93.3	10.1	89.6	11.1
		male	78	84.3		80.1	
16	Isolated vessel tortuosity	female	112	75.3	8.8	65.8	14.8
		male	111	82.2		76.3	
17	Laser spots	female	141	97.0	2.9	94.3	1.9
		male	156	94.3		92.5	
18	Lens dislocation	female	4	100	0	100	6.1
		male	7	100		94.0	
19	Maculopathy	female	66	92.2	7.0	88.4	8.3
		male	81	98.9		96.1	
20	MH	female	31	96.4	8.8	95.5	27.2

		male	17	88.3		72.6	
21	Myelinated nerve fiber	female	9	100	0	98.9	35.8
		male	4	100		68.9	
22	Normal	female	304	96.2	0.6	95.0	1.8
		male	208	96.8		96.8	
23	Optic abnormalities	female	66	70.7	0.4	62.0	13.9
		male	98	70.4		71.2	
24	PDR	female	122	97.9	0.9	97.7	0.6
		male	186	97.0		97.1	
25	Peripheral retinal degeneration	female	457	93.9	7.6	94.3	9.3
		male	265	87.1		85.9	
26	PM	female	155	96.4	1.3	97.3	3.0
		male	80	97.7		94.4	
27	RD	female	61	93.9	1.7	89.8	1.8
		male	118	92.3		88.3	
28	Retinal breaks	female	27	96.4	7.6	94.7	18.5
		male	43	89.3		78.7	
29	Retinal white dots	female	8	100	none	92.1	none
		male	0	none		none	
30	RP	female	24	100	0	99.8	0.2
		male	19	100		100	
31	Silicone oil	female	24	100	0.06	100	1.4
		male	41	99.9		98.6	
32	Surgery-air	female	16	99.6	0.4	100	0
		male	14	100		100	
33	Surgery-band:buckle	female	9	100	20.1	100	13.1
		male	8	81.7		87.7	
34	Surgery-medicine	female	2	58.3	39.3	75	18.1
		male	15	86.9		90.0	
35	TRD	female	28	83.2	4.0	81.8	0.5
		male	46	80.0		81.3	
36	Vasculitis	female	14	100	10.2	98.2	11
		male	40	90.3		88.0	
37	Vitreous opacity	female	109	87.9	7.2	84.4	6.3
		male	177	94.5		89.9	
38	VKH	female	40	99.2	1.3	98.5	0.7
		male	47	98.0		97.8	

AH, asteroid hyalosis; BRVO, Branch Retinal Vein Occlusion; CRVO, Central Retinal Vein Occlusion; ERM, Epiretinal Membrane; FEVR, Familial Exudative Vitreoretinopathy; HM, High Myopia; MH, Macular Hole; PDR, Proliferative Diabetic Retinopathy; PM, Pathological Myopia; RD, Retinal Detachment; RP, Retinitis Pigmentosa; TRD, Tractional Retinal Detachment; VKH, Vogt-Koyanagi-Harada Disease.

6) There is no information about the train/val/test split process or size of groups, or hyperparameter tuning, if any

Response: Thanks for this good suggestion. Tables R10 and R11 demonstrate the details of model training hyperparameters and the data size for training and testing, respectively. For a fair

comparison, all approaches adopt their official model settings and the same data settings. The implementation details have been revised in the new manuscript as follows.

“We use similar network configurations for the experiments with the OculoScope and MixNAF datasets. The parameters of the ResNet-101 network are initialized from the model pre-trained on the ImageNet. Common data augmentation methods are used to enrich the data. For instance, we scale the size of input fundus images to 512 x 648 and augment the set of training data with random horizontal and vertical flips. The details of training, validation, and test dataset division are shown in Supplementary Table 8. For the OculoScope and MixNAF datasets, we carry out multi-class multi-disease classification and perform fairness analysis. The network is optimized using the Adam optimizer. The initial learning rate is 1e-4, and a decaying learning rate is implemented by using OneCycleLR for each batch with a weight decay=1e-4, betas=(0.9, 0.999). We trained the network for 100 epochs with a batch size of 16. The entire model is built on a Ubuntu 18.04 system with PyTorch and two NVIDIA GeForce RTX 2080Ti. The loss weight parameters of λ_1 , λ_2 , λ_3 , and λ_4 are experimentally set to 0.01, 1, 0.01, and 1e-5, respectively.”

Table R10. Demonstration of the model training hyperparameters.

Dataset	OculoScope	MixNAF
Optimizer	Adam, weight_decay=1e-4, betas=(0.9, 0.999)	
Scheduler	OneCycleLR	
Learning Rate	1e-4	
Epochs	100	
Batch size	16	
GPUs	2080Ti × 2	
λ_1	0.01	
λ_2	1	
λ_3	0.01	
λ_4	0.00001	

Table R11. The division of training and test datasets.

Dataset	Training set	Testing set
OculoScope	13,855	2,675
MixNAF	3,614	926

7) There is no complete list of abbreviations for diseases. The methods sections lists a few followed by “etc”. Abbreviations for retinal diseases are used throughout the tables and figures without a listing of all abbreviations

Response: Good suggestion. The complete list of abbreviations for retinal diseases involved is demonstrated in Table R12. This table is also added to the revised manuscript as Supplementary Table 10.

Table R12. Complete list of abbreviations for retinal diseases involved. A total of 41 diseases are included in the ultra-widefield and narrow-angle datasets.

ID	Abbreviation	Full name
1	AMD	Age-Related Macular Degeneration
2	CHRP	Congenital Hypertrophy of the Retinal Pigmented Epithelium
3	HM	High Myopia
4	PPA	Peripapillary Atrophy
5	BRAO	Branch Retinal Artery Occlusion

6	BRVO	Branch Retinal Vein Occlusion
7	ARN	Acute Retinal Necrosis
8	CMV	Cytomegalovirus Retinitis
9	MFC	Multifocal Choroiditis and Panuveitis
10	TB	Tuberculosis
11	VKH	Vogt-Koyanagi-Harada Syndrome
12	PM	Pathological Myopia
13	NV	Neovascularization
14	RD	Retinal Detachment
15	CRAO	Central Retinal Artery Occlusion
16	CRVO	Central Retinal Vein Occlusion
17	CSC	Central Serous Chorioretinopathy
18	DR	Diabetic Retinopathy
19	FEVR	Familial Exudative Vitreoretinopathy
20	MOCD	Melanocytoma of the optic disk
21	RB	Retinoblastoma
22	UM	Uveal Melanoma
23	VHL	Von Hippel-Lindau Disease
24	VPTR	Vasoproliferative Tumors of the Retina
25	IRDs	Inherited Retinal Diseases
26	BCD	Best's Disease
27	RP	Retinitis Pigmentosa
28	STGD	Stargardt Disease
29	PCV	Polypoidal Choroidal Vasculopathy
30	CNV	Choroidal Neovascularization
31	EALES	Eales Disease
32	FBRV	Frosted Branch Retinal Vasculitis
33	IRVAN	Idiopathic Retinal Vasculitis, Aneurysms, and Neuroretinitis
34	AH	Asteroid Hyalosis
35	ERM	Epiretinal Membrane
36	MH	Macular Hole
37	IOL	Intraocular Lens
38	TRD	Tractional Retinal Detachment
39	VD	Vitreous Disease
40	PDR	Proliferative Diabetic Retinopathy
41	PS	Posterior staphyloma

8) The paper would benefit from additional English language editing. While the writing is generally well-organized, there are a places where the specific wording is strange/unusual/not easy to understand

Response: Good suggestion. Based on your suggestion, we turned to Nature Author Gold Services (verification code: 8769-B7CB-1872-492B-217P) to improve the presentation of the manuscript (Figure R2). We also carefully revised the paper based on your suggestions in terms of the text and presentation.

Figure R2. Manuscript editing certificate at Springer Nature Author Services.

9) Methods, lines 53-54: “we calculate the average of the above metrics for each disease to measure the overall screening performance of a model.” – is this a micro-average, so weighted for number of photos for each disease? Or macro-average?

Response: Good point. It is calculated using macro average. We have revised it in the new manuscript to make it clearer.

“we calculate the macro-average of the above metrics for each disease to measure the overall screening performance of a model.”

10) There are a few diseases which caused some specific concerns, like “cataract” or “vitreous opacity” – presumably one could only diagnose “media opacity” – how could you tell that the blurry image was due to cataract vs corneal opacity vs vitreous opacity? This seems strange to me. Also, what is “surgery-medicine”? also “Maculopathy” is pretty general – does this include age-related macular degeneration? (it must also include other diseases since it seems that some young people had maculopathy)

Response: We greatly appreciate the good concerns. We first demonstrate the quantitative experimental results for cataract, vitreous opacity, and media opacity, as shown in Table R13. Next, we Response to your questions point-by-point.

Table R13. Quantitative experimental results for cataract, vitreous opacity, and media opacity.

Disease	Method	Screening accuracy metric				Fairness metric (age 10 years)			
		AP (%)	Spec.	Sens.	AUC	$\Delta D \downarrow$ (%)	PQD \uparrow (%)	DPM \uparrow (%)	EOM \uparrow (%)
Cataract	Baseline	80.1	0.876	1.0	0.949	89.1	80.4	5.1	88.5
	FairerOPTH	87.9	0.934	0.906	0.972	32.0	86.8	21.4	84.6
Vitreous opacity	Baseline	87.9	0.908	0.941	0.977	109.0	94.4	12.1	86.4
	FairerOPTH	92.1	0.922	0.958	0.986	86.1	93.7	32.4	94.7
Media opacity	Baseline	85.0	0.858	0.896	0.944	71.8	81.1	26.9	71.4
	FairerOPTH	89.2	0.867	0.937	0.961	27.1	88.8	27.3	85.7

(1) **Clarification on Disease Identification.** We appreciate the reviewer’s keen observation regarding specific diseases such as “cataract” and “vitreous opacity” in our annotation process.

It is indeed important to clarify how we differentiate between these conditions. For "cataract," the diagnosis is based on a thorough assessment of the patient's wide-field fundus photos. In cases of cataract (Figure R3a), the entire retina in the patient's fundus photos exhibits a consistent, homogeneous turbidity, without any obvious signs of turbidity in the vitreous cavity, such as bleeding or turbid vitreous masses. This diagnosis is further supported by a review of the patient's medical records and the attending doctor's diagnosis, allowing us to confidently label the condition as "cataract" in the photos.

Figure R3a. Typical cataract shown in our UWF images dataset

Figure R3b. Typical vitreous opacity shown in our UWF images dataset

Figure R3c. Typical "Surgery-Medicine" shown in our UWF images dataset

Regarding "vitreous opacity" (Figure R3b), our wide-angle photos offer a distinct advantage as they clearly depict turbid vitreous masses within the vitreous cavity. This clarity in displaying vitreous opacities is a notable strength of wide-angle photos and is often challenging to achieve with narrow-angle photos. According to the above two characteristics, some pictures are labeled with both cataract and vitreous opacity. It is important to note that our photo library excluded images of corneal opacity or partial cataract based on the clinical data provided by patients and their healthcare providers.

- (2) **Explanation of "Surgery-Medicine"**. The term "surgery-medicine" refers to the presence of strips or powdery, clearly defined, white lumps within the patients' vitreous (Figure R3c). These lumps are typically the result of treatments involving medications such as triamcinolone acetonide or OZURDEX, which are administered to address macular edema.
- (3) **Comprehensive Nature of "Maculopathy"**. In our FairerOPHTH AI system, "Maculopathy" encompasses a range of conditions, including age-related macular degeneration (AMD), Polypoidal Choroidal Vasculopathy (PCV), and other macular lesions such as hemorrhage and

atrophy. We opted for a broader category due to the limitations of wide-angle fundus photos in accurately displaying fine macular details, especially in comparison to narrow-angle photos or images obtained using specialized equipment like the ZEISS CLARUS 500 Fundus Camera. Our choice to use the term "maculopathy" is rooted in the aim of raising awareness among patients and doctors regarding potential macular issues. It serves as an alert to consider further, more detailed examinations, recognizing the challenge of diagnosing specific conditions like AMD or PCV solely based on single wide-field fundus photos.

In summary, our disease identification process is rigorous, and we rely on a combination of wide-angle fundus photos, clinical data, and healthcare provider diagnoses to categorize and label conditions accurately. We appreciate the reviewer's valuable input, which has enabled us to provide additional clarity on our methodology and system categories.

11) A limitation is that fairness according to the sensitive attributes of race/ethnicity could not be performed, presumably as their ultra-wide-field dataset was not collected on patients with diverse race/ethnicity

Response: We sincerely appreciate the reviewer's discerning observation regarding the absence of fairness considerations based on the sensitive attribute of race/ethnicity in our study.

Demographics of our patient population. Our hospital is situated in Shanghai, a region located in East China. The majority of our patients come from the surrounding provinces and the local area. In this geographical context, the Han nationality constitutes the overwhelming majority of our patient population. As a result, our dataset primarily comprises patients of Han ethnicity.

Limitation to be included in the article. We recognize the importance of considering and addressing potential disparities related to race and ethnicity in healthcare research. In light of the reviewer's comment, we explicitly mention this limitation in the revised manuscript to ensure transparency and clarity about the demographic composition of our patient dataset.

The limitation added to the discussion Section is described below.

"We recognize the importance of considering and addressing potential disparities related to race and ethnicity in healthcare research. However, a limitation is that the demographics of our patient population can not perform the fairness evaluation to the sensitive attributes of race/ethnicity. Our hospital is situated in Shanghai, a region located in East China. The majority of our patients come from the surrounding provinces and the local area. In this geographical context, the Han nationality constitutes the overwhelming majority of our patient population. As a result, our dataset primarily comprises patients of Han ethnicity."

12) Were there normal fundus photos among set? Of course in any "screening" settings the baseline population would be mostly normal so it's not clear if there are no normal images in the set whether the performance would do well when those types of images are included.

Response: Yes. There are normal fundus images in both the ultra-widefield and narrow-angle datasets. Table R14 reports the detailed number of images. This has been clarified in the revised version.

Table R14. Demonstration of the number of normal fundus images in the training and testing datasets.

Dataset	Training set	Testing set
OculoScope (ultra-widefield)	2,423	512
MixNAF (narrow-angle)	1,622	365

13) When comparing screening accuracy of oculoscope to other models (MCAR, CSRA, C-Tran, etc.) there is no information in the methods about how the other models were implemented, hyperparameter tuning, etc.

Response: Good suggestion. To conduct a fair comparison, the implementation of compared models such as MCAR^[4], CSRA^[5], C-Tran^[6], ML-Decoder^[7], and ASL^[8] adopts the official code and hyperparameter settings provided by the authors of the paper. The code for these models is available on the corresponding authors' GitHub². For hyperparameter configuration, please see the author's description in the paper.

14) While the FairerOphth model contains a classification branch for retinal pathological features, performance on this was not shown. Similarly, I wonder if it's possible that there could be predictions in the two branches that are contradictory, i.e. classifying the features incorrectly but the diagnosis correct, or vice versa. If so that would certainly limit the "explainability" advantage of this approach

Response: We greatly appreciate the good suggestion. We first demonstrate the classification performance for the retinal pathological features on the OculoScope and MixNAF datasets, as shown in Table R15. Our FairerOPHTH achieves 0.951 and 0.950 AUC on the OculoScope and MixNAF datasets, respectively. Even the disease and pathological features are classified with high accuracy, we agree with you that there may be a contradiction between the predictions of the two branches. This issue has been taken into account in our approach. We used the joint loss term L_{joint} to join the GT labels of pathological features and diseases together to form a new label "pathology + disease", and then used the ASL loss^[8]. The L_{joint} is used to enhance that the classified pathological features and diseases are consistency. Compared with previous models that cannot automatically provide the retinal pathological features corresponding to classified diseases, the proposed approach has made significant progress in interpretability.

Following common model interpretation method using class activation maps (CAM)^[9], we further demonstrate the CAM results of FairerOPHTH (Figure R4). Based on the suggestion, we have added this discussion and limitation to the revised manuscript.

Table R15. Classification performance for the retinal pathological features on the OculoScope and MixNAF datasets. Accurately classifying the 67 categories of retinal pathological features in OculoScope and the 20 categories in MixNAF is greatly challenging. Our FairerOPHTH achieves decent classification performance.

Retinal pathological features	Sensitivity	Specificity	AUC
OculoScope (ultra-widefield)	0.939	0.911	0.951
MixNAF (narrow-angle)	0.910	0.927	0.948

Figure R4. Visualization of class activation maps (CAM)^[9] of FairerOPHTH.

² <https://github.com/gaobb/MCAR>; <https://github.com/Kevinz-code/CSRA>; <https://github.com/QData/C-Tran>; https://github.com/Alibaba-MIIL/ML_Decoder; <https://github.com/Alibaba-MIIL/ASL>.

Reference

- [1] Du, S., Hers, B., Bayasi, N., Hamarneh, G., & Garbi, R. (2022, October). FairDisCo: Fairer AI in dermatology via disentanglement contrastive learning. In European Conference on Computer Vision (pp. 185-202).
- [2] Feldman M, Friedler S A, Moeller J, et al. Certifying and removing disparate impact." proceedings of the 21th ACM SIGKDD international conference on knowledge discovery and data mining. 2015: 259-268.
- [3] Hardt, Moritz, Eric Price, and Nati Srebro. "Equality of opportunity in supervised learning." Advances in neural information processing systems 29 (2016).
- [4] Gao, Bin-Bin and Hong-Yu Zhou. "Learning to Discover Multi-Class Attentional Regions for Multi-Label Image Recognition." IEEE Transactions on Image Processing 30 (2020): 5920-5932.
- [5] Zhu, Ke and Jianxin Wu. "Residual Attention: A Simple but Effective Method for Multi-Label Recognition." 2021 IEEE/CVF International Conference on Computer Vision (ICCV) (2021): 184-193.
- [6] Lanchantin, J., Wang, T., Ordonez, V., & Qi, Y. (2020). General Multi-label Image Classification with Transformers. 2021 IEEE/CVF Conference on Computer Vision and Pattern Recognition (CVPR), 16473-16483.
- [7] Ridnik, T., Sharir, G., Ben-Cohen, A., Ben-Baruch, E., & Noy, A. (2021). ML-Decoder: Scalable and Versatile Classification Head. 2023 IEEE/CVF Winter Conference on Applications of Computer Vision (WACV), 32-41.
- [8] Baruch, E.B., Ridnik, T., Zamir, N., Noy, A., Friedman, I., Protter, M., & Zelnik-Manor, L. (2020). Asymmetric Loss For Multi-Label Classification. 2021 IEEE/CVF International Conference on Computer Vision (ICCV), 82-91.
- [9] Zhou, B., Khosla, A., Lapedriza, À., Oliva, A., & Torralba, A. (2015). Learning Deep Features for Discriminative Localization. 2016 IEEE Conference on Computer Vision and Pattern Recognition (CVPR), 2921-2929.

Reviewers' Comments:

Reviewer #1:

Remarks to the Author:

The revised manuscript has effectively addressed all of my previous concerns. I recommend its publication without any further modifications.

Reviewer #2:

Remarks to the Author:

The authors have made many additional analyses available which have improved transparency in the reporting of the results.

1) Regarding generalizability of model: We appreciate the clarification that the model does not need pathological feature annotation in order to be deployed "tested" on fundus images. However, the limitation still exists that if any other user were to try to use this model to adapt to their particular task, and needed to fine-tune the weights to their dataset, they would not be able to do so without pathological annotations. I think this could be acknowledged.

2) The analyses on adapting for other modalities of data - this is now I think really confusing because it's not clear what kind of text is used as an input and how it was generated and such. I appreciate the author's efforts here but probably it would be better to either leave this out or actually put in some more clarifying detail.

3) The authors have addressed my other comments well.

Response to Reviewers

Manuscript #NCOMMS-23-33455A

We express our gratitude to all the reviewers for their constructive comments, which have been immensely helpful in improving our manuscript. We have carefully addressed all the reviewers' points. We are confident that the new manuscript will address all the concerns raised by the reviewers.

The reviewers' comments are in black font, while our responses are in blue font.

REVIEWERS' COMMENTS

Reviewer #1 (Remarks to the Author):

The revised manuscript has effectively addressed all of my previous concerns. I recommend its publication without any further modifications.

Reply: Thank you for your thorough review and constructive feedback.

Reviewer #2 (Remarks to the Author):

The authors have made many additional analyses available which have improved transparency in the reporting of the results.

1) Regarding generalizability of model: We appreciate the clarification that the model does not need pathological feature annotation in order to be deployed "tested" on fundus images. However, the limitation still exists that if any other user were to try to use this model to adapt to their particular task, and needed to fine-tune the weights to their dataset, they would not be able to do so without pathological annotations. I think this could be acknowledged.

Reply: Thank you for your valuable feedback. We agree with you that users would need to provide pathological annotations if they are to retrain or fine-tune the model for their particular tasks. We added this point in the revised manuscript.

2) The analyses on adapting for other modalities of data - this is now I think really confusing because it's not clear what kind of text is used as an input and how it was generated and such. I appreciate the author's efforts here but probably it would be better to either leave this out or actually put in some more clarifying detail.

Reply: Thanks. We are sorry for the confusion and appreciate your constructive suggestion. The input text is a description of fundus features in the fundus image to evaluate that other types of data such as commonly used text can be adapted to our architecture. Based on your comments, we have further clarified the input text in the revised manuscript.

3) The authors have addressed my other comments well.

Reply: Thank you for your thorough review and constructive feedback.

Reviewer #2 (Remarks on code availability):

I did visit the github and verified that there is code available. Have not performed a complete attempt at reproducing but at least on cursory overview it looks appropriate. Of note the first google drive link given in the manuscript (where oculoscope and mix-naf datasets are) appears to be locked "Access Denied"

Reply: Thank you. We have opened access to Google Drive.